# The Irradiation Effects in Ferritic, Ferritic–Martensitic and Austenitic Oxide Dispersion Strengthened Alloys: A Review

**DOI:** 10.3390/ma17143409

**Published:** 2024-07-10

**Authors:** Natália Luptáková, Jiří Svoboda, Denisa Bártková, Adam Weiser, Antonín Dlouhý

**Affiliations:** Institute of Physics of Materials, Czech Academy of Sciences, Žižkova 22, 616 62 Brno, Czech Republic; svobj@ipm.cz (J.S.); bartkova@ipm.cz (D.B.); weiser@ipm.cz (A.W.); dlouhy@ipm.cz (A.D.)

**Keywords:** nuclear materials, ferritic oxide-dispersion strengthened steel, ferritic–martensitic oxide dispersion strengthened steel, austenitic oxide-dispersion strengthened steel

## Abstract

High-performance structural materials (HPSMs) are needed for the successful and safe design of fission and fusion reactors. Their operation is associated with unprecedented fluxes of high-energy neutrons and thermomechanical loadings. In fission reactors, HPSMs are used, e.g., for fuel claddings, core internal structural components and reactor pressure vessels. Even stronger requirements are expected for fourth-generation supercritical water fission reactors, with a particular focus on the HPSM’s corrosion resistance. The first wall and blanket structural materials in fusion reactors are subjected not only to high energy neutron irradiation, but also to strong mechanical, heat and electromagnetic loadings. This paper presents a historical and state-of-the-art summary focused on the properties and application potential of irradiation-resistant alloys predominantly strengthened by an oxide dispersion. These alloys are categorized according to their matrix as ferritic, ferritic–martensitic and austenitic. Low void swelling, high-temperature He embrittlement, thermal and irradiation hardening and creep are typical phenomena most usually studied in ferritic and ferritic martensitic oxide dispersion strengthened (ODS) alloys. In contrast, austenitic ODS alloys exhibit an increased corrosion and oxidation resistance and a higher creep resistance at elevated temperatures. This is why the advantages and drawbacks of each matrix-type ODS are discussed in this paper.

## 1. Introduction

In nuclear fission, energy is gained by splitting heavy atoms such as U^235^ and Pu^239^ into lighter atoms. This process is triggered when an atom’s nucleus absorbs a neutron, which makes the nucleus unstable and leads to a release of 2–3 neutrons per one split up event. The multiple neutrons released in the event enable a subsequent chain reaction. In contrast, nuclear fusion is a process by which atomic nuclei of a low atomic number join and form a heavier nucleus with lower energy while the excess energy is released. As a typical example, in the core of the Sun, hydrogen nuclei fuse to produce helium nuclei, but this is an extremely demanding reaction and cannot be used for applications on Earth. For practical fusion energy production, a reaction between two heavy isotopes of hydrogen, deuterium and tritium seems to be the closest achievable alternative.

Engineering of parts used in nuclear applications has proven challenging and often the selection of viable material requires the parts to have a particular combination of physical, chemical and nuclear properties [1]. While the nuclear-related aspects are extremely challenging, a fusion thermomechanical environment is even more problematic. Materials used in the first wall of a reactor are exposed to transient heat waves associated with plasma disruptions. These deposit gigawatts of thermal energy into very small regions of the first wall in a very short time. Parts required to transfer heat are also required to have more mass and have generally more complex shapes compared to typical fast reactor fuel cladding components. Stresses may also arise from different thermal expansion coefficients and temperature gradients that occur over distances of several meters. These stresses and strains must be accommodated by means of interconnected structures maintaining precise dimensional tolerances. In addition, the thermo-mechanics are complicated by spatially varying dimensional instabilities due to the swelling, creep and evolution of material properties. In conclusion, we can state that the exceedingly demanding thermomechanical environment of fusion process is unparalleled and proves to be a grand challenge with the potential for large-scale energy production. The material issues on the first wall of fusion reactors have not yet been sufficiently addressed [2]. Structural materials exposed to radiation doses are damaged by displacement. Transmutation products give rise to complex changes in macroscopic dimensional stability, mechanical properties, and corrosion resistance. These changes require a modification of the material microstructures from atomic to mesoscopic-length scales in order to suppress the effects of irradiation exposure and consider the impact of a large number of mechanisms and variables acting in combination over long periods of time [3]. The development of appropriate structural and functional materials for applications in fission and fusion reactors represents an enormous challenge for materials scientists.

The objective of this paper is to create an overview of the progress achieved in the research on ferritic and austenitic oxide dispersion strengthened (ODS) alloys as structural components for current and future nuclear reactors. It is general knowledge that structural steel udergoes complex changes in precipitate composition as well as the growth of helium bubbles and microvoids under neutron irradiation. These changes are connected to drops in the thermo-mechanical properties. Moreover, currently used fuel cladding materials are reexamined due to their limited tolerance to service accidents. In general, available austenitic, ferritic and ferritic–martensitic steel matrices (furthermore referred to as FM steels) can serve as the base for the corresponding ODS material. Each FM steel has its advantages/disadvantages and the properties of the final ODS alloy significantly depends on the chemical composition and processing of the final alloy. Up until now, most of the research has targeted ODS alloys with ferritic and/or FM matrices, and the feasibility of austenitic ODS alloys has received only limited attention [4,5]. This is probably due to the fact that the mechanical alloying (MA) of austenitic ODS alloys is more difficult in comparison to ferritic ODS alloys [6]. On the other hand, it must be kept in mind that the stability of the microstructure in FM ODS alloys is guaranteed only at temperatures below the ferrite-to-austenite transformation, which significantly limits their applicability.

First-generation nuclear reactors were generally built out of former state-of-the-art high-performance alloys. The selection was impaired by limited knowledge regarding the radiation-induced, thermomechanical and chemical degradation processes. In the following decades, a deeper understanding of radiation effects and other degradation processes in materials has been achieved. The proper choice of structural materials, e.g., for fuel cladding, structural components and reactor pressure vessels, is vital for the safety and economics of current and proposed nuclear fission and fusion energy systems [7].

As the research on ODS alloys has progressed significantly in the last decade, it is impossible to discuss all the results published in the literature in a single review paper. Thus, only the following topics are covered by this review: (i) stainless steels for nuclear applications; (ii) a brief history of ODS steels in the context of fission reactors; (iii) ferritic and FM ODS alloys and (iv) austenitic ODS alloys. 

## 2. Stainless Steels for Nuclear Applications

Designs of new-generation nuclear reactors and thermal power plants are based on the research regarding the mechanical behavior of FM and austenitic steels subjected to fatigue and/or creep at elevated temperatures (450–650 °C). The thermomechanical and radiation degradation of materials is primarily influenced by the operation. High-temperature creep must be considered for operating temperatures exceeding 0.5 TM (TM is the melting temperature) and may be significant even at operating temperatures in the range of ~0.35–0.4 TM. A relation between temperature and the radiation damage is, furthermore, complicated by the potential presence of multiple radiation effects.

All steels, including stainless steels, consist of iron with the addition of carbon. The alloy will be called steel when carbon improves the strength by a precipitation of carbides. In the opposite case, we will talk about an alloy. The steel can exhibit different crystalline (lattice) structures, which depend on the chemical composition and processing conditions. Ferrite, austenite and martensite are possible lattice modifications, and all are found within different types of steels. Austenite crystallizes in face-centered cubic (fcc) lattices, is non-magnetic and forms different variants of stainless steel. The austenitic steels usually exhibit a very low stacking fault energy, a high work-hardening capability and a high ductility of several tens of percentages. Certain alloying elements, most notably nickel, stabilize the austenite such that austenite-to-ferrite or austenite-to-martensite transformations do not occur during cooling to room temperature. High chromium ferritic or FM steels have a number of advantages in comparison to austenitic stainless steels employed in the early fission reactors. In particular, FM steels exhibit lower thermal expansion, better void swelling resistance and higher thermal conductivity. 

Swelling by definition is the isotropic volume expansion of a material, without application of external stresses. This is caused by two phenomena. The Frenkel pairs of vacancies and interstitials are created by irradiation. Supersaturation due to vacancies leads to nucleation of cavities known as void swelling. Moreover, helium produced by (n, α) reactions condensates in the cavities and creates a high pressure environment, which is known as bubble swelling [8]. Garner [9] shows that the steady-state swelling rates of iron-based alloys are not very different. The difference in swelling is primarily influenced by the crystal lattice’s structure, chemical composition, deformation rate, microchemical inhomogeneities and temperature history determining the duration of the transient regime of swelling. The peak swelling temperature was mentioned in several studies. These studies indicated that the swelling in FM steels peaked at around 420–470 °C in ion irradiations [10] and neutron irradiations [9] and is affected by the damage rate linked to vacancy generation. However, helium may reduce the mobility of vacancies and increase the stability of voids. Over temperatures of 470 °C, helium effects are expected to be enhanced. Therefore, the targeted irradiation temperatures range from up to 700 °C for helium effects in ODS ferritic steels. The maximum swelling rate was observed at 500–600 °C [8] because of the increase in helium mobility at such temperatures [11]. In conclusion, the peak swelling is determined as a result of a balance between the mobility of vacancies, production of helium and thermal stability of voids [12].

High chromium ferritic and FM steels, typically containing 9–12 wt.% Cr, have been developed since the 1930s [13]. In the 1970s, high-chromium (9–12% Cr) FM steels became candidates for high-temperature applications in the core of fast reactors. Since the 1970s, advances have been made in developing steels with 2–12% Cr for conventional power plants that significantly improved the efficiency of energy production [14]. Calculations showed that important alloying elements in conventional FM steels, particularly Nb and Mo with long activity decays after irradiation, must be replaced by metallurgically equivalent Ta and W [15,16]. Also, concentrations of Ni, Cu, Al, Co and some other impurity elements with a long-term activity must be drastically reduced [17]. 

Since the mid-1980s, several experimental FM steels with reduced activation (RAFM steels) have been created [18,19,20]. The International Energy Agency (IEA) set up international collaborations to accelerate the development of RAFM steels in 1992 [21]. The high-temperature performance of current RAFM steels concerning creep and other mechanical loadings is comparable to that of “second-generation” FM steels [22]. Many countries have developed their own RAFM steels, such as EUROFER97 [23] in Europe, F82H [24] and JLF-1 [25] in Japan, 9Cr–2WVTa [26] in the USA, ARAA [27] alloy in South Korea, CLAM [28] in China and Italy developed the BATMAN RAFM steels [29]. A Brief history of RAFM steel for nuclear applications is shown in Figure 1. 

The main elements contained in RAFM steels are C, Cr, Mn, W, V, and Ta. C and Mn are austenite-promoting elements, and Cr, W, V, and Ta are ferrite-promoting elements. The chemical compositions, heat treatment and microstructure of typical RAFM steels are mentioned in Table 1 [30]. These steels have carefully balanced amounts of alloying elements such as Cr, W, Ta, and V [31] and show an improved void swelling even under high irradiation doses [17]. Moreover, the amount of C and N needs to be adjusted with caution in order to achieve proper strengthening by secondary precipitates. The radiation resistance of materials is affected by the matrix chemical composition as well as by the type of precipitates. Bcc steels exhibit higher void swelling resistance compared to fcc steels due to the reduced relaxation volume of bcc lattices [32]. This is caused by differences in solute trapping and in the character of the dislocation loop microstructures [9,33]. Furthermore, the number density and size of M_23_C_6_-type carbides must be controlled as coarsening of this phase may worsen the impact properties [34]. The RAFM steels serve in quenched as well as tempered conditions and profit from both the martensitic microstructure and higher ductility. However, peak temperatures during service may lead to undesired austenitization and subsequent martensite formation upon cooling [35]. 

RAFM steels typically contain 8–9% Cr and 1–2% W. The solid solution of W and Cr causes a certain degree of lattice distortion in the matrix, which hinders the dislocation slipping, resulting in improved strength. Many dislocations are developed during the normalizing process of RAFM steel. Even though high-temperature tempering significantly reduces the dislocation density in the steel, usually 10^14^ m^−2^ dislocations remain in the tempered RAFM steel. Dislocations cut and anchor each other during the deformation process, which serves as dislocation strengthening. Precipitation strengthening is the most important strengthening method for RAFM steels. Many M_23_C_6_ (M = Cr/Fe) and MX (M = V/Ta, X = C/N) phases precipitate during tempering. This is important for improving the high-temperature microstructural stability as well as mechanical properties of RAFM steel, because the precipitates effectively pin grain boundaries and hinder dislocation motion. Numerous interfaces are observed in RAFM steel after normalizing and tempering, such as the original austenite grain boundaries, tempered martensite lath boundaries, lath block boundaries, lath bundle boundaries and the boundaries between the second phases and matrix [22,37,38]. Figure 2 shows the schematic diagram of the microstructure of typical RAFM steel after tempering with differently oriented lath bundles distributed in the original austenite grains. The lath bundle includes multiple high-density dislocations, M23C6 on the lath boundaries and MX in the bulk.

RAFM steels are also considered for fuel cladding applications, with a focus on safe containment of the fuel and radioactive fission products and simultaneous efficiency in the heat transfer from the fuel to the coolant [7]. The typical heat flux through the cladding material is about 1 MW/m^2^ (about 1% of the heat flux of the surface of the Sun). The decay time of common elements in RAFM steel is shown in Table 2 [30]. The fuel service life is typically several years, during which the cladding material is continuously exposed to high temperatures (300–400 °C), mechanical stress and intense irradiation. Further requirements on the fuel cladding material include oxidation and corrosion resistance, and a low rate of hydrogen absorption, thus limiting hydrogen cracking issues [30,39].

RAFM steels are also presently considered as materials of choice for the first thermonuclear fusion power reactor ITER (International Thermonuclear Experimental Reactor) and the demonstration fusion power plant (DEMO) [14,40]. However, there have been discussions concerning their applicability in the fusion program, in particular due to issues associated with the application of a ferromagnetic material in high magnetic fields. The ferromagnetic material could cause magnetic field perturbations with detrimental consequences for plasma stability, and, in turn, the high external magnetic fields of the reactor could generate forces destroying the steel. Moreover, the applicability of RAFM steels in the harsh neutron radiation environment of a fusion plant has also been questioned [14]. Additional details concerning the metallurgy of FM and RAFM steels can be found in the excellent monograph by Klueh and Harries [41].

Specific fusion-related steel development programs have thus been initiated. For instance, in Europe the “MANET” steel (Martensitic steel for the Next European Torus) with improved creep, impact and toughness properties resulted from the optimization of commercial alloys [20,42]. As compared to standard ferrite microstructures, the tempered martensite usually exhibits a smaller micrograin size and has a denser and more complex system of internal boundaries which serve as efficient sinks for point defects created by irradiation. Such microstructures provide additional resistance to void nucleation. Very interesting results were reported regarding the effect of δ-ferrite on the impact properties [43], the possibility to substitute carbon with nitrogen and the often-observed creep instability. All these findings enabled scientists to specify and procure a new European reference alloy called MANET II (10Cr0.6MoVNb) [20]. When tested in neutron and heavy-ion fluxes [44], the ferrite phase showed significantly higher resistance to swelling as compared to the FM phases [21].

The effects on precipitates in CLAM steel irradiated at 500 °C with 792 MeV Ar ions was studied by Shen et al. [45]. The densities of vacancies nearest to MC particles were significantly higher than those in the bulk at low radiation doses. This indicates a preferential generation of vacancies at the interface between the second phase and the matrix. Contrarily, vacancies and inert atoms are captured by the MC interfaces which play a crucial role in suppressing the high-temperature expansion of the martensite lath boundary [30]. These effects are demonstrated in Figure 3, showing that cavities preferentially nucleate at the interface of MC precipitates under different radiation doses.

Tan and et al. [37] developed castable nanostructured alloys (CNAs) as a new generation of RAFM steels to overcome their certain limitations. A class of CNAs captures the positive engineering of RAFM and NFA/ODS alloys. Grade 91 has comparable tensile properties in the same manner as current versions of CNAs such as F82H and Eurofer97 [46]. The CNAs contain a significantly increased amount of MX nanoprecipitates, leading to superior properties embracing the merits of current RAFM steels and NFA/ODS alloys. Nanoscale precipitates improve the radiation resistance of CNA steels [46] and adequate precipitate stability was reported for irradiation up to ~50 dpa [37,47].

Terentyev [48] developed RAFM steel for nuclear applications with a reduced Mn, Si and C content. This newly developed low Mn/C steel demonstrated that an unconventional heat treatment allows the production of a material with a transition temperature as low as −133 °C with a tensile strength comparable to that of Eurofer97. On the flip side, the performed quench and rolling procedure does not lead to any sizeable reduction in the DBTT. There is a significant increase in the tensile strength without any compromise to the ductility in the relevant operational temperature range.

Despite the new types of RAFM steels such as CNA, CLAM, SIMP, Ti-RAFM, ARAA, IN-RAFM etc. the FM/RAFM steels for fusion applications are far from technological maturity and their qualification for service is thus limited. The strength and corrosion requirements limit the applicability of FM steels to temperatures below about 550 °C. While this may be unjustifiably optimistic for service in neutron environments, where data available for higher temperatures, longer times (10^4^ h are typical maximum times in creep tests) and a combination of creep and fatigue are scarce, dimensional instabilities such as void swelling (at high helium levels) represent an issue near and above 400 °C, where synergistic irradiation–thermal instabilities are possible. At temperatures below 350 °C degradation of toughness and loss of ductility are limiting, while an embrittlement at high He levels acts at a much wider range of service temperatures [49]. On the contrary, in a recent study [50], the authors investigated the effect of oxide particles on the evolution of He bubbles and dislocation loops on CLAM steel and FM ODS steel. Helium bubbles were not observed inside FM ODS steel grains or at grain boundaries. This fact suggests that the oxide particle interface is more capable of capturing helium atoms compared to other sink mechanism. The limits can be attributed to the significantly lower amount of nanoprecipitates in the current RAFM steels [46]. Such low amount is insufficient to pin the lath boundaries during recovery processes at temperatures above 600–650 °C.

The severe loss of uniform elongation due to irradiation seems to be the major problem with all FM and RAFM steels. Moreover, cavity swelling scenarios in RAFM steels are not fully documented. Thus, enhanced swelling may occur under the influence of the fusion neutron spectrum. Then, dimensional instabilities, further worsened by anticipated swelling–creep–fatigue interactions and synergistically aided by He/H generation beyond >20–50 dpa, can reach unacceptable values. The performance of RAFM steels in the intermediate temperature range (~400–500 °C) is also not fully evaluated because only a few fusion-relevant neutron irradiation experiments have been performed up to now. Thus, it is rather complicated to determine the acceptable lifetime doses being affected by cavity swelling as well as by other degradation mechanisms [51].

Introducing ODS methodology is expected to generally improve the situation in that it would increase the range of operating temperatures while maintaining the advantages of the FM and RAFM steels concerning the irradiation effects. Employing oxide dispersion in FM steels may yield additional swelling resistance, first by pinning the grain walls and thus maintaining the micrograin’s small size and second by allowing the dispersoid interfaces to serve as sinks for point defects. Building on this idea, new ODS ferritic or FM steels produced by powder metallurgy processes have been designed. They contain high densities of nanosized oxides the interfaces of which were also shown to trap gasses, thereby mitigating embrittlement effects due to helium. With these promising properties, advanced ODS steels can be considered as materials for use in the fusion environment [21].

## 3. A Brief History of ODS Steels in the Context of Fission Reactors

Before we summarize the development of ODS alloys in the past, it is first necessary to mention the history of fission reactors. The fission reactors used as part of commercial services today belong predominantly to Generation II, built in the 1960s and 1970s, and a part of this family is already operating in a lifetime extension mode [52,53,54]. New Generations III and III+ fission reactors, with more advanced design, are starting to be deployed. An extensive research effort is currently taking speed in order to develop the Generation IV nuclear reactors. As compared to the older generations, their design should guarantee improved safety, reliability, sustainability, longer lifetimes, proliferation resistance and profitability [55].

It was reported in a Nature paper in 1967 [56] that voids are formed in austenitic grains during irradiation. Development of materials for fuel elements in fast reactors profited from electron microscopy investigations of thin stainless steel foils irradiated in the Dounreay Fast Reactor. The samples were obtained from either experimental fuel elements or a bulk irradiated material intended for subsequent mechanical testing. Shortly thereafter, this detrimental feature of austenite-based steels was used by Huet as one of the arguments in favor of ODS ferritic alloys [57,58,59]. Consequently, the very first ODS iron-based alloy was prepared in 1967 [57]. The base matrix material was obtained in the form of pre-alloyed powder (AISI 410 or 405) or by mixing powdered Fe and Cr. This caused the early attempts using a direct mix of metallic and oxide powders to be arduous to produce. The investigated oxides were Al_2_O_3_, MgO, ZrO_2_, TiO_2_ and ZrSiO_4_. However, the best preliminary results were obtained with TiO_2_, which led to very good mechanical properties at high temperatures [60]. Yttria (Y_2_O_3_) powder was first considered in 1970 when it was added to the matrix at a concentration of 1 wt.% and the mechanical alloying yielded very good results. The addition of yttria motivated the biggest development in the field of ODS alloys during the following years [57]. More advanced development took place in Japan during the 1990s by the Power Reactor and Nuclear Fuel Development Corp and concerned ODS with 11–12% Cr and 9% Cr [61,62,63,64]. The latter efforts focused on the relations between concentration levels of Y_2_O_3_, Ti and excess O and the oxide dispersion parameters. Attention was also given to the martensitic transformation in the ODS 9%Cr alloys aimed at a suppression of the grain anisotropy that typically develops in ferritic 12% Cr ODS alloys during hot consolidation. Early studies showed Ti to be beneficial for decreasing the grain size and increasing the particle density of Y-Ti oxide particles. In addition, the early ODS alloys were considered when thinking of how to improve fuel cladding durability in the advanced fast reactors [65]. In about the year 2000, the discovery of a high-number density of Y-Ti oxide nanoparticles or even nanoclusters (NC) in 12YWT helped to increase interest in the development of ODS alloys for nuclear applications [66]. Following this discovery, atom probe tomography (APT) confirmed Y-Ti-O NCs in MA957 [67]. Concurrently with these discoveries, development of 14YWT ODS alloys within a framework of US DOE programs started in 2000. This advanced ODS alloy, also referred to as nanostructured ferritic alloy (NFA), targeted a high point-defect sink density via a high number density (about 5–10 × 10^23^ m^−3^) of 2–5 nm sized nanoclusters and micrograins with mean size below 500 nm that provide a very high interfacial surface area [68,69,70,71]. In many cases NFAs are classified by their base chromium content ≈ 14Cr and microalloys Y, W and Ti. A typical example of NFA composition is (wt.%) 14Cr, 1–3 W (or 0.75–1.5Mo), 0.2–1Ti, 0.15–0.3Y, 0.1O, bal. Fe. The atoms of Y, Ti and O are completely dissolved by a combination of gas atomization of powders and by mechanical alloying [72]. In particular, the NFA variants exhibit a combination of high mechanical strength concerning tensile, creep and fatigue loadings, coupled with a high resistance to the irradiation [73,74]. A further development of 14YWT took place in the Advanced Fuel Campaign (AFC) which was done under US DOE’s Nuclear Technology Research and Development (NTRD) Program. NTRD investigated fabrication methods for the production of thin wall fuel cladding. Joining operations by friction stir welding and understanding the effects of neutron irradiation on the microstructure and mechanical properties were also a part of the research program. The NTRD program aimed to develop a high-performance fuel cladding in sodium-cooled fast reactors (SFR). Dependable fuel integrity and proper fuel behaviour is of utmost importance in SFR or lead-cooled fast reactors (LFR), which require a very high-temperature performance [75]. Improved resistance to avoid swelling and other microstructural degradation processes occurring at radiation damage levels up to 150 dpa or higher and superior creep properties at temperatures up to 700 °C are a given requirement for alloys used in such reactors [76]. Numerous studies demonstrated that the ODS program goals have been achieved. The strategy changed a bit after the 2011 nuclear accident in Japan, where interest has been given to accident-tolerant fuel cladding for light water reactors in order to increase the safety of the current technology [77]. For these reasons, a basic accident scenario [78] considered a temperature range corresponding to steam oxidation at 1000–1500 °C. Under these conditions, ODS Fe-15Cr alloys cannot form a protective scale while FeCrAl alloys offer low oxidation rates at even higher temperatures of 1400–1500 °C [79].

Up to now, rather limited results have been reported for FM ODS alloys. A 9Cr FM ODS alloy was shown to offer a promising swelling resistance and improved strength [80,81,82]. Over the past few decades, MA technology has been optimized to enable the formation of oxide dispersion in the alloy matrices that is as fine and homogeneous as possible. Such dispersions effectively prevent grain coarsening and swelling during operation in nuclear power plants [83,84,85,86,87]. Based on the optimized composition of ODS ferritic steels, thin-walled cladding tubes were manufactured using the improved MA process, hot extrusion and hot working. In 1989 a first cladding was produced in cooperation with a fabrication vendor. The mechanical properties (i.e., tensile, uni- and bi-axial creep rupture and Charpy impact properties) were extensively examined on the manufactured cladding tubes [88]. Among others, the traditional fusion welding generally deteriorate ODS properties due to changes in grain size and modifications of the nano-oxide distribution [89,90]. Consequently, solid-state welding processes were preferred and tested. One of the welding processes developed for a nuclear-grade ODS steel is resistance upset welding. Joule heating associated with a compressive mechanical load at the plug-to-clad contact is sufficient to ensure a proper weld joint [91]. The fabrication routes of cladding tubes have also been described for ferrite-type ODS steels involving recrystallization and for a FM-type one utilizing γ to α phase transformation. A joining process between cladding tube and the end-plug has been developed based on the pressurized resistance upset welding method [88].

France’s nuclear program considers two main ODS families as viable, the Fe-9Cr martensitic ODS alloys and the Fe-14/18Cr ferritic ODS alloys. The martensitic steels with their phase transformation ranging above 800–850 °C are usually more isotropic and easier to manufacture. The downside is the lower corrosion resistance compared to high chromium ferritic ODS alloys. ODS alloys are more anisotropic, more difficult to manufacture and are likely more susceptible to embrittlement under irradiation. The final choice between candidate materials mentioned above has not been made yet. General studies continue and are focusing on fabrication options, mechanical properties before, under and after irradiation, compatibility with fuel and Na and behavior during fuel reprocessing. The final choice will be based on previous results obtained on the first generation of ODS alloys irradiated in different reactors [92].

Two 13% Cr ODS ferritic alloys reinforced by yttria and titania, developed by SCK/CEN Mol [57] and elaborated on by DOUR Metal as fuel pin claddings (Fetiche experiment), have been irradiated in the experimental subassemblies of Phénix (called IDEFIX). The pin claddings from the Fetiche experiment were characterized in the 1990s [93]. The dose reached was 81 dpa at temperatures ranging from 400 to 580 °C. The results indicated a significant worsening of mechanical properties after neutron irradiation. Hardening was due to α′ and χ-precipitates and formation of dislocation loops induced by irradiation [93]. The severe embrittlement of the irradiated ODS cladding, rupture and longitudinal cracks were associated with the presence of oxide-free strips parallel to the tube axis in the as-received material state, see Figure 4a. These microstructural irregularities induced a heterogeneous dislocation loop distribution after irradiation and accompanying localization of plasticity, see Figure 4b [92].

As was already mentioned, the high flux of 14 MeV neutrons in fusion reactors produces substantially more He and H as compared to similar radiations doses imposed in fission reactions. This is related to a larger proportion of high-energy neutrons in their energy spectrum characteristic for the fusion environment. We note that the neutron capture cross-sections for a generation of He and H in (*n*, α) and (*n*, p) reactions are significant at neutron energies that are above those where the fission neutron spectra fall to low values. Helium is essentially insoluble in solids and this is ultimately the reason for the bubble formation and associated detrimental effects. He embrittlement of steels has been attracting much attention since the mid-1960s, when Barnes [94] attributed the high-temperature embrittlement of irradiated steels to helium bubbles in grain boundaries. A popular introduction to the subject was given by Ullmaier in 1982 [95] on the occasion of an International Symposium on “Fundamental Aspects of Helium in Metals”. Accumulation of Helium at grain boundaries is known to compromise the integrity of the fusion materials. Helium can significantly degrade the tensile, creep and fatigue properties at elevated temperatures. Helium bubble at grain boundaries may lead to crack initiation and premature failure under stress. The extent of degradation depends on temperature, He concentration, He production rate, stress level and the composition and microstructure of the material [3].

Elongation properties of ODS steels can be direction-dependent, depending on the production process. This leads to better properties measured along the extrusion direction while inferior properties are measured in the other directions [51]. Requirements for material properties vary depending on the wall thickness of the part. Fracture toughness is the most important in the first wall blanket steel structures, which tend to be much more complicated and thick-walled. On the other hand, for thin-walled fast reactor cladding tubes, fracture toughness is not of paramount importance. Higher tensile ductility before and after neutron irradiation does not automatically mean better properties. ODS steels generally have worse fracture toughness compared to tempered martensitic FM steels. Fracture toughness does not correlate to tensile ductility, but depends more on hardness (σ_YS_)—most hard materials typically show poor fracture toughness irrespective of their tensile ductility. It should also be noted that the irradiated tensile ductility of other alloys after high-dose neutron irradiations was not quantified in the literature [51].

At present, in addition to the developments in the framework of nuclear fission programs, intense research at a global level has been conducted in the field of nuclear fusion technology. However, the enormous complexity in building fusion systems and the associated immense material challenges remain [96]. The history and development of ODS steels is presented in Figure 5.

## 4. Ferritic and FM ODS Alloys

Ferritic matrix-based ODS steels are classified into two categories: (i) ferritic and (ii) FM ODS. The alloy design in the former aims for a high concentration of Cr exceeding 12 wt.%, which offers an advantage in terms of corrosion resistance. The Cr concentration in the FM ODS steels is limited to 12 wt.%. The FM ODS steels are expected to have better radiation resistance due to a high concentration of sink sites for irradiation defects represented by both oxide/matrix interfaces and fine-grained and high dislocation-density microstructures. In addition, they have better hot workability, which is due to softening caused by the ferrite-to-austenite phase transformation [97]. In the development of FM ODS steels for nuclear applications, a Cr concentration of 8–9 wt.% is preferable. This alloy design suppresses the ductility loss by irradiation hardening and improves the microstructure stability and creep strength at high temperatures. Conversely, from the viewpoint of corrosion resistance, Cr content at a level of 11–12 wt.% Cr is preferable [98]. Henry [99] manufactured several 11–12Cr FM and duplex ODS steels and, as a result of the increased Cr content, observed excessive residual α-ferrite which did not transform during austenitization. Moreover, 11Cr-ODS steel containing a small portion of residual α-ferrite was successfully manufactured by adding element-stabilizing austenite. This 11Cr-ODS steel had satisfactory creep strength and ductility comparable with those of the 9Cr-ODS steel [51].

At present, there are numerous commercial and experimental ODS alloys, which are often based on FM steel, see Table 3. According to the chemical composition of the ODS alloys, two groups can be identified. First, titanium alloyed ODS steels (e.g., 14YWT) are typically characterized by the dispersion of fine (<10 nm) [67] oxide particles in the matrix, leading to enhanced strength. Second, aluminum alloyed ODS grades (e.g., PM2000) contain larger oxide particles (<50 nm) [100]. They exhibit higher corrosion resistance [101,102], especially in aqueous media [103] and oxidation resistance in air atmospheres up to temperatures of 1450 °C [104]. However, their mechanical strength is less spectacular [105,106]. Several developed ODS steels exhibit outstanding properties, and this section briefly describes some of the exceptional alloys.

ODS Eurofer is the result of the European research effort regarding reduced activation ferritic/martensitic (RAFM) alloys [118]. Experimental alloys 12Y1, 12YWT and 14YWT were produced in Japan by Sumitomo Industries and Kobe Special Tube, respectively. Fe-18Cr1W was investigated in collaboration with A&D and Plansee with consolidation performed at CEA/SRMA at a semi-industrial scale [108]. Kanthal APMT^TM^ steel was investigated for possible use as an accident-tolerant fuel cladding material in advanced light water reactors [119]. The MA 957 was manufactured by INCO Metals and MA 956 is a product of Special Metals Corporation. PM 2000 is commercial product of Metallwerk Plansee GMbH in Germany. Another ODS alloys is C26M, a second-generation FeCrAl alloy for nuclear-grade enhanced accident-tolerant fuel (ATF) cladding developed by Oak Ridge National Laboratory (ORNL) [120]. Tempered martensitic ODS steels with 9 and 18 wt.% chromium (9Cr-, 18Cr-ODS steel) were developed by the Japan Atomic Energy Agency (JAEA).

The typical microstructures of ODS steels include sub-micron grain size, defect sink densities near or greater than 10^16^ m^−2^, nano-oxides with average diameters <*d*> ≈ 5 nm, number density *N* ≈ 10^23 − 24^ m^−3^ and volume fraction 0.5–1%, coherent and semi-coherent interface relationships with misfit dislocations between the nano-oxides and matrix [121,122,123,124].

Most structural alloys are produced by casting followed by thermomechanical treatment provoking hardening due to precipitation of, e.g., Mg_2_Si, CuAl_2_, ZrAl_3_, Mg_2_Al_3_ in Al alloys [125,126,127] or to carbides or nitrides in steels [128]. This is possible, because, e.g., in steels a sufficient amount of C or N can be dissolved in their solid state at a high temperature and the precipitation is provoked by lowering the temperature, often supported by *γ* to *α* phase transformation. Such alloys can be produced in large amounts by proven technologies. This is unfortunately not the case for ODS alloys. It is not possible to dissolve a sufficient amount of O in the liquid or solid alloy as the oxides are chemically stable up to very high temperatures [129]. The ODS alloys can be produced only by utilizing the powder metallurgy consisting of the following steps (see Figure 6).

1. The ODS powder is produced by mechanical alloying (MA) of the input powders or granulates. During the MA the balls collide and small powder particles are forged onto the balls to form protrusions. Large protrusions break off as large particles, which are gradually broken into small ones. These processes are repeated and lead to powder homogenization. In several studies, such as [130], it is demonstrated that Y_2_O_3_ powder particles are only refined and dispersed in the matrix during the MA. Now, it is accepted that Y_2_O_3_ can be completely dissolved in the matrix during the MA and the O and Y atoms are trapped at drastically multiplied defects as dislocations and vacancies [125,131] and/or form clusters [132]. Y_2_O_3_ re-precipitation is then conditioned by a significant decrease in defect density.

2. The canned ODS powder can be consolidated by spark plasma sintering [133,134,135,136], hot isostatic pressing (HIP) [137,138] or hot forming as rolling [139,140,141], pressing [142,143], hot extrusion [144], rotary swaging [145,146], forging or by their combinations [147,148,149]. All these methods of consolidation and subsequent thermomechanical treatments are quite complicated and expensive and represent the key processing step.

3. Spark plasma sintering or hot isostatic pressing may lead to already completed stable microstructures with fine grains strengthened by oxide nanodispersion [150]. After consolidation by hot forming methods applied at significantly lower temperatures, the microstructure usually consists of much finer grains with very fine oxide dispersion and still a huge density of defects. Such an ODS alloy is hard and brittle and may contain sufficient stored energy to drive secondary recrystallization.

4. Secondary recrystallization can take place if the driving force overcomes the pinning of grain boundaries by nanodispersion. To achieve a coarse-grained microstructure, the driving force cannot be too large, so that only a small number of grains with the best conditions overcome the pinning effect and grow to large sizes [151].

Many research groups are working on the powder metallurgical fabrication of Y_2_O_3_-based ODS steels that includes mechanical alloying and subsequent consolidation via HIP, hot extrusion or spark plasma sintering. However, the fundamental mechanisms of the fabrication steps are not fully understood. It has been observed that during mechanical alloying changes in the structural properties of Y_2_O_3_ take place. After milling for several hours, the intensities of the Y_2_O_3_ peaks in powder X-ray and neutron diffraction (XRD) patterns decrease and finally disappear [64,152,153,154]. After consolidation or heat treatment of the milled powders, Y_2_O_3_ or, if the matrix contains Ti or Al, a Y-(Ti/Al) complex oxide reappears [155,156]. Similar phenomena have been observed for different secondary phases such as YFe_3_ and Fe_2_O_3_ [157].

Since the material performance of ODS steels depends on the morphology of oxide particle dispersion, dispersion control is essential for the development of the steel [12]. The Al-addition to FeCr-ODS steels altered the main particles from Y_2_Ti_2_O_7_ pyrochlore to YAH and YAP [123], and a small addition of Zr to FeCrAl ODS steels altered the majority of the oxide particles from Y-Al-O dispersoids to Y-Zr-O dispersoids [158], resulting in the enhanced high-temperature strength of the steel [101].

Chauhan and coworkers [159] compared properties of ODS alloys with different microstructures and they focused on an ODS Fe-9%Cr-based ferritic alloy and an ODS Fe-9%Cr-based FM ODS alloy. The pre-alloyed powders were produced by gas atomization. Mechanical alloying of the pre-alloyed powders with Y_2_O_3_ particles was performed under a hydrogen atmosphere within a vertical attributor ball mill. The milled powders were then sealed in soft steel cans, followed by degassing and were subsequently hot extruded into bars at 1100 °C. Thereafter, the ODS Fe9Cr steel bar was annealed at 1150 °C for 1.5 h and a dedicated heat treatment was performed to establish the FM microstructure of the tube material [159]. TEM bright field micrographs of ODS Fe9Cr after extrusion as well as after the heat treatment are shown in Figure 7.

After the hot extrusion, the steel exhibits equiaxed grains with a low dislocation density (0.46 ± 0.1) × 10^14^ m^−2^. The dark contrast Cr carbides are present both at grain boundaries and within the grains. Irregularly shaped Cr-W enriched carbides and Ti-enriched particles are located at grain boundaries. In both cases, complex Y–Ti–O nanoparticles are nonuniformly distributed within the matrix. Nevertheless, regions with a uniform oxide distribution were also observed, see Figure 8 as an example. Therefore, it can be concluded that the heat treatment did not change the size and distribution of the nano-oxides.

Meza et. al. [116] have developed a new 14Cr ODS steel by MA followed by spark plasma sintering (SPS) consolidation of the 14Cr pre-alloyed powder with complex oxide formers such as Y-Ti-Zr and different alloying elements like B. Boron is known as the microstructural stabilizer due to its pinning effect on grain boundaries. The use of the precursors resulted in a better control of the nano-oxide composition in ferritic ODS microstructures. A required precipitation morphology of the nano-oxides was achieved, which guaranteed a strong pinning effect of the dislocations and grain boundaries. The selected parameters in the SPS technique fully densified the MA powder mixture while avoiding an excessive grain growth and maintaining a high density of dislocations during the consolidation process. The required parameters were achieved without further heat or thermo-mechanical treatments. The subsequent mechanical testing performed at room and high temperatures (500 °C) showed very promising results, comparable to the ones obtained for ODS steels developed with a more complex processing route [116].

In the temperature regime relevant for GenIV application (T ≈ 650 °C < 0.4 Tm), the Fe-14Cr alloys can be loaded to rather high stresses before dislocation creep mechanisms get activated. Hence, the grain refinement caused by the oxide dispersion will not deteriorate the creep behavior. The ODS steels developed as SFR fuel cladding material contain at most 12% chromium. It is well known that the corrosion resistance in high-temperature water diminishes significantly as the chromium content decreases below 13% [160]. The ODS Fe-14Cr-2W-0.3Ti alloy was investigated by Šćepanović et al. [161] before and after high-temperature triple-ion beam and low-temperature single-ion beam irradiation. Simultaneous triple-ion beam irradiation experiments with Fe^5+^, He^+^ and H^+^ ions were performed to simulate fusion damage on two nanostructured ferritic alloys with nominal composition Fe-14Cr-0.3Y_2_O_3_ and Fe-14Cr-2W-0.3Ti-0.3Y_2_O_3_. The study revealed the development of nanovoids or small bubbles, which were not present in the virgin samples, even though the dispersion showed virtual compositional stability. The nanoparticles maintained their composition and morphology after the triple-ion beam irradiation, while their sizes slightly increased [161].

The Eurofer ODS steel belongs to a very well-known commercial grade of the ODS materials. This ODS steel was developed within the framework of the European Fusion Technology Programme [162]. Eurofer ODS steel was simultaneously prepared using powder metallurgy methods at the Centre of Research in Plasma Physics (CRPP, Switzerland) [82] and at the CEA Grenoble, France [162]. Concerning its thermal stability, it was found that within the ferritic phase field (below 850 °C), the 9%Cr ODS Eurofer steel exhibits only a small amount of softening [163,164]. This is due to reinforcement of the steel with oxide nanoparticles, which are able to hinder dislocation and grain boundary migration and trap radiation-induced defects. Eurofer97 produced by gas atomization of a pre-alloy has a microstructure formed by martensite laths. The input oxide powders usually consist of crystallites with average sizes ranging from 20 to 100 nm [165]. The martensite laths formed in the pre-alloyed powders transform into approximately equiaxed grains with a high density of dislocations (~10^16^ m^−2^) during MA [166,167]. After MA the powders exhibit a very fine grains with an average grain size of tens of nanometers while some grains host oxide particles [168]. Generally, after MA the ODS steel powders exhibit a significant refinement of their microstructure which depends on the fraction of the added oxide particles and on their chemical nature [165]. In comparison with MgAl_2_O_4_, addition of Y_2_O_3_ leads to a finer microstructure. A higher content of Y_2_O_3_ causes a smaller grain size. Additionally, Cayron et al. [165] concluded that, while Y_2_O_3_ dissolves in the steel matrix during the mechanical alloying, the MgAl_2_O_4_ particles do not show this ability. Other authors [64,135,154,166,169] have also found that Y_2_O_3_ dissolves progressively in the steel matrix and reprecipitates as nanosized clusters during hot consolidation. Zilnyk [35] compared two Eurofer steels, (i) Eurofer-97 and (ii) ODS-Eurofer. The grain microstructure of both of them are shown in Figure 9. Eurofer-97 contains M_23_C_6_ carbides rich in Cr, Fe and W within a wide size range (30 nm to 300 μm) and MX (TaC and VN) carbonitride particles with sizes less than 50 nm. Similarly, the ODS-Eurofer steel also hosts the M_23_C_6_ carbides, but MX-type particles are not present in this material due to a consumption of Ta and V by the Y_2_O_3_ nanoparticles. Concerning the Eurofer-97 steel, an important finding is that its final grain size increases with austenitizing temperature. Above 1000 °C, when M_23_C_6_ and MX particles start to dissolve, the grain size shifts to significantly larger values. On the other hand, in ODS-Eurofer steel the grain size is rather insensitive to the increasing austenitizing temperature since the strong pinning effect is imposed on grain boundaries by the finely dispersed Y-based nano-oxides [35].

Klimenkov [170] pointed out that the occurrence of radiation-induced defects depends also on the local number density and the size distribution of the oxide particles and not only on the irradiation damage and temperature [171]. The ODS-Eurofer97 behaves almost identical to RAFM steels, where the uniform elongation reduces severely to values below ~0.5–1% for doses as low as ~1 dpa. Thereafter, the uniform elongation saturates at the lower values and is almost non-existent [51]. Kolluri et al. conducted neutron irradiations on Eurofer-ODS steel to 1–3 dpa at 300 °C–550 °C [172,173]. They found that high-density black spot defects appeared at low temperatures, and low-density but large size dislocation loops and dislocation networks were observed at high irradiation temperatures. After neutron irradiation of Eurofer 97 and Eurofer-ODS up to about 16 dpa, the average size of dislocation loops in Eurofer 97 is 7 nm at 250 °C, while the high-density oxide particles region in Eurofer-ODS steel has very few dislocation loops [174].

A commercial ODS alloy PM 2000 was developed and produced by the International Nickel Company (INCO) and later it was offered by the Plansee Company in Austria. PM2000 is 20% Cr-ODS steel containing 5% Al, which exhibits superior resistance to oxidation and corrosion in hot gases at temperatures > 1000 °C. The advantage of PM2000 is that the Al addition imparts better corrosion and oxidation resistance due to a durable passivating Al_2_O_3_ surface scale. The scale can withstand service conditions in lead–bismuth eutectic cooled advanced fission reactors. This alloy also involves ~0.2 wt.% Y_2_O_3_, but during thermomechanical processing this oxide phase is converted to yield a much different distribution and morphology of particles than found in NFAs. Mechanical alloying of the starting powders and their hot consolidation leads to a dispersion of Y-Al-O oxide precipitates with an average diameter of 20–30 nm at a density of ~10^21^ m^−3^, being considerably coarser than nanostructured ferritic alloys (NFA) [175]. The alloy can also be considered for accident-tolerant fuel cladding for commercial light water reactors [78,176,177]. However, this class of ODS ferritic stainless steels needs further systematic characterization, especially under conditions of intensive irradiation producing He atoms [178]. Limited studies of the microstructural response of PM2000 to self-ion and α particle irradiation have been reported in [155,179,180,181]. Such irradiations indicate that the Y-Al-O precipitates are very stable under charged particle irradiation, however, they can trap helium that leads to void and bubble formation on the matrix/particle interfaces. One of these studies also found evidence of a strong association between helium bubbles and dislocation loops formed when PM2000 was implanted with α particles under uniaxial stress [182]. The results of the study by Jung [178] have shown that PM2000 undergoes substantial changes that might significantly degrade its performance. These changes, particularly the chemical segregation, high density of dislocation loops and radiation-induced precipitation, may alter both the corrosion resistance and the mechanical properties of PM2000, especially at higher levels of helium dpa.

Using atom probe tomography, Miller and coworkers [183] investigated the formation, nature and stability of the nanometer-sized particles during high-temperature annealing treatments close to the melting point of the molybdenum-containing alloy MA957. The INCO’s MA957 steel was described in 1978 by a US Patent issued to the INCO Company [184]. As compared to the MA957, a similar tungsten-containing MA ODS ferritic alloy designated as 12YWT exhibited remarkable improvement in creep properties at temperatures up to 850 °C [62,63]. The intragranular volume of the 12YWT alloy contains a high number density of ultrafine Ti-, Y- and O-enriched particles that were resistant to coarsening during high-temperature creep or isothermal annealing at up to 1300 °C [66,185,186].

Miller et al. [187] also pointed out that MA ODS ferritic alloys are attractive for fusion reactor applications because of their potential to withstand high temperatures and the ability of dispersed oxide particles to trap developing helium. The microstructural stability of MA/ODS Fe-14 wt.% Cr-3% W-0.4% Ti-0.3% Y_2_O_3_ ferritic alloy was also investigated by the atom probe tomography. The study provided evidence that MA/ODS 14YWT ferritic alloys contain a high density of nanosized titanium-, yttrium- and oxygen-enriched clusters in the as-extruded condition. These nanoclusters were found to be extremely resistant to coarsening at temperatures of at least 1000 °C. However, the atom probe tomography maps also revealed that chromium with relatively low affinity to oxygen participated in the formation of oxides. This is evidently due to the lack of yttrium and titanium in the system. The increase in their content can further increase the resistance of the oxide dispersion to coarsening.

In the year 2009 Sakasegawa et al. investigated a MA957 ODS steel (Fe-14Cr) and proposed a correlation between the size of the nano-oxides and their chemical composition [188]. The smallest precipitates (2 to 15 nm in diameter) were identified as non-stoichiometric Y-Ti-O, the particles in a size range 15 to 35 nm were stoichiometric Y_2_Ti_2_O_7_ sized oxides, whereas bigger ones corresponded to Ti-oxides. Miller et al. reported similar results for a 14YWT alloy where compositions Y_2_Ti_2_O_7_ and Y_2_TiO_5_ accounted for 5 to 10 nm precipitates and non-stoichiometric Y-Ti-O for 1 to 4 nm precipitates [189]. One reason for these stoichiometry-sized correlations could be that the non-stoichiometric Y-Ti-O clusters are metastable and observed just at the stage of their dissolution. Nevertheless, recent studies tend to demonstrate that very small nano-oxides in a size range of 2 nm and less, already possess the Y_2_Ti_2_O_7_ or Y_2_TiO_5_ stoichiometry [190,191]. After MA, Spartacus [192] reported clusters composed of Y, O and Ti which nucleated and grew together with a sequential enrichment in Ti (up to 700 °C) and Y (between 900 and 1100 °C). At 1100 °C the nano-oxides exhibited the composition Y_2_Ti_2_O_7_ which changed to Y_2_TiO_5_ during further isothermal holding. In this state, the nano-oxides display an extremely low coarsening rate.

Based on their maturity, the ODS alloys can be divided into either already commercialized or under development categories. Concerning the second group, Kim et. al. [107] observed the finest and homogeneously distributed oxide particles in the 12YWT alloy and reported on its highest high-temperature tensile and creep rupture strengths. These results were explained by the formation of complex oxides of Ti–Y–O due to addition of Ti. Also, W atoms show homogeneous distribution in the matrix, suggesting a solid solution-strengthening effect. The fine complex oxides of Ti–Y–O interfered with grain growth and increased the thermal stability of the matrix during high-temperature exposures, see Figure 10.

Edmondson et al. pointed out that nano-scale oxides could trap helium atoms and thus promote a formation of ultra-fine helium bubbles with a high density in a He-implanted 14YWT [193]. On the other hand, Lu et al. found that higher density and finer nano-oxide particles are able to better suppress bubble formation in He-implanted 14Cr-ODS steels [73]. By comparing ODS with non-ODS steels, the remarkable irradiation damage tolerance of ODS variants has also been extensively discussed in many previous publications.

According to Song [12], the excellent swelling resistance is dominantly attributed to the high sink strength of oxide particles that depends on the morphology of particle dispersion rather than the crystal lattice structure of the particles. In contrast, no dislocation loops were produced in any of the irradiated steels. Nanoindentation measurements showed that no irradiation hardening but softening was found in the ODS ferritic steels, which was probably due to irradiation-induced dislocation recovery. The helium bubbles in a high density never contributed to the irradiation hardening of the ODS steels in these irradiation conditions.

The observation that irradiation-induced hardening decreases with the increasing density of oxide particles is obviously a common feature of ferritic–martensitic 9–10Cr ODS steels (like Eurofer-ODS) and ferritic 13–17Cr ODS steels (like 14YWT). In the case of ODS steels, the density of nano-scaled ODS particles (e.g., Y_2_O_3_, Y_2_Ti_2_O_5_) significantly exceeds the dislocation loop density observed in the conventionally cast ODS particle-free steels irradiated under the same conditions. For higher than critical density of oxide particles, irradiation-induced Frenkel defects likely do not contribute to the nucleation and growth of stable dislocation loops, but rather annihilate by trapping it at the ODS–particle matrix interface. This is why the uniformly distributed nano-scaled ODS particles have an outstanding additional function: they are not only the cornerstone for the high-temperature creep and fatigue strength, but they also significantly reduce irradiation hardening and thus retard irradiation embrittlement at low strain rates relevant for applications [174].

As mentioned earlier, the microstructure of ODS ferritic steels consolidated via extrusion consists of large, elongated grains and nanosized grains. In addition, the material has a strong <110> α-fibre texture, formed during the extrusion process. The anisotropy in grain morphology and the <110> α-fibre texture lead to poor mechanical properties in the transverse direction [147]. Additional thermomechanical processing is often applied to ensure the elimination of porosity and enhance their performance. Recently, hot-cross rolling was used by Auger [194] to improve the resistance of 14YWT ODS alloy against radiation. Zhang et al. [195] applied hot rolling to increase the strength of Fe-9Cr-0.06C-1.5W-0.5Ti-0.18Si-0.35Y_2_O_3_ ODS alloy (wt.% are used in all notations). Kumar et al. [147] studied the mechanical properties of Fe-18Cr-2W-0.2Ti-xY_2_O_3_ alloy consolidated from mechanically alloyed powder by forging. Zhou et al. [196] applied combinations of forging, hot rolling, cold rolling, and annealing to ODS310 alloy with Mo addition. However, inhomogeneous grain size distribution and brittle cracking along the grain boundaries are still issues for the ODS alloys [197]. Moghadasi et al. [198] produced an ODS ferritic alloy with nanometric Y_2_TiO_5_ particles dispersed in the matrix via a vacuum casting route. Although the coarse grain microstructure achieved by completed secondary recrystallization seems to be the key factor for acquiring excellent creep strength, several ODS alloys presented in the available literature have not met this requirement yet [197]. Just finding a simple processing route leading reliably to the coarse-grained microstructure seems to be the key task in the research on ODS alloys.

The newest ODS alloy is GRX-810, developed by NASA [199]. The GRX-810 (Glenn Research Center eXtreme) alloy was developed and is being further matured to allow an increase in operating temperatures for these various components. The alloy is Ni–Co–Cr-based and incorporates ODS coating of the powder to allow for high performance at elevated temperatures. Traditionally manufactured ODS alloys were plagued with high expenses due to the manufacturing process [200], but additive manufacturing is enabling more economical high-performance ODS alloys, like GRX-810, to be produced. This also enables high-complexity ODS alloy components such as injectors.

A new research strategy based on the design of coarse-grained (CG) ferritic ODS alloys seems to yield alloy variants with excellent creep and oxidation resistance up to 1300 °C. Such materials may also find their application in nuclear industries. Svoboda et al. [129] developed a new CG ferritic ODS nanocomposite Fe-10Al-4Cr-4Y_2_O_3_ (in wt.%) (referred to as FeAlOY in what follows), strengthened with 5 vol.% of Y_2_O_3_ 20 nm sized nano-dispersoid. The MA powders are subjected to hot consolidation by rolling or rotary swaging after which the FeAlOY alloy exhibits an ultra-fine-grained (UFG) microstructure. The CG variant is subsequently formed by secondary recrystallization. One of the major characteristics of the hot consolidated FeAlOY is that the microstructure consists of very fine equiaxed grains (typical size range between 100–200 nm, see Figure 11) characterized by a high internally stored energy, high hardness and brittleness. The recrystallization treatment is typically performed at 1200 °C, leading to the formation of a very coarse, highly anisotropic grain microstructure. After the recrystallization, the creep resistance of the CG FeAlOY is improved enormously, qualifying the alloy for high-temperature applications [201]. One of the most impressive properties of the CG FeAlOY is the existence of a rather high threshold stress under which the alloy exhibits only short primary creep and later the creep rates drop to nearly unmeasurable values. This property of the CG FeAlOY guaranties the shape stability of the parts (e.g., pulling rods of the mechanical testing machines) loaded at temperatures up to 1300 °C. Recent experiments revealed the limited cohesive strength of grain boundaries being the weakest element concerning the strength of the CG FeAlOY. This could motivate a further development aimed at a complete elimination of grain boundaries. A prospective production of single-crystalline (SX) FeAlOY alloy variants may lead to further improvement of the creep strengths.

The cohesive strength of grain boundaries in the FeAlOY alloy can be stimulated by appropriate alloying elements. Nearly 20 different metallic elements have been tested with the objective of improving the parameter. Moreover, some researchers have also experimented with the small additions of yttrium–oxide (Y_2_O_3_), typically coupled with Ti, Zr, Mg, Hf and/or Al, into a wide variety of Fe and Ni-based materials, all aimed at improving ultimate tensile strength and creep resistance at high temperatures during irradiation [72,74,165,203,204]. A positive outcome was obtained for Al, Cr and Y, whereby the influence of Y is enormous (addition of 1% of metallic Y increases the strength by a factor of 2). Results by Svoboda et al. [205] presented in Figure 12 suggest that the tensile strength of the FeAlOY alloy increases with increasing Y content in both Fe-10Al-4Y_2_O_3_ and Fe-10Al-4Cr-4Y_2_O_3_ ODS variants up to about 1.2% of yttrium after which it drops to low values. This drop in strength is accompanied by a significant increase in ductility due to the existence of a large volume fraction of non-recrystallized ultra-fine-grained regions. Just this microstructure exhibits at this temperature a rather low creep strength due to diffusional creep and ductility. On the opposite end, in the case of high strength, low ductility is observed caused by an intergranular brittle fracture, which can be accounted for by the grain boundary decohesion. Thus, the strength of the specimens can serve as a measure of the cohesive strength of the grain boundaries, provided the grain microstructure of the specimens is similar.

Results of the tensile tests performed at 1100 °C and strain rate of 10^−6^ s^−1^ and presented in [205] showed that the amount of added metallic Y substantially influences the strength of the ODS alloys. The authors concluded that the significant increase in the strength is due to changes in the grain morphology, see Figure 13.

The most important property for any structural material expected to operate at elevated temperatures is high-temperature creep. Figure 14 presents a Larson–Miller diagram that compares the FeAlOY with other conventional ODS, superalloys and single-crystal materials. The advantage of the diagram consists in the fact that it allows to compare the creep properties of different materials by using only one parameter. In this figure, the stress to rupture is plotted against the Larson–Miller paramater (LMP), which is defined as LPM = *T*(20 + log*t*), where *T* is the temperature in Kelvin and *t* is the rupture time in hours. The more valuable the creep properties of a material, the nearer the corresponding curve lays to the upper-right corner of the plot.

Compared to classic ODS alloys, the FeAlOY contains a ten-times higher volume fraction of the stable Y_2_O_3_ nanodispersion, which gives the alloy its high-temperature strength. Furthermore, the high content of Al in the matrix guarantees excellent oxidation resistance. Above 1100 °C the FeAlOY reaches the ultimate tensile strength of 100 MPa and plasticity of 1%. To demonstrate the interaction of dislocations with the nano-oxides in the FeAlOY at a strain rate of 1 × 10^−6^ s^−1^ and temperature 400 °C, STEM micrographs are presented in Figure 15. The majority of the Y_2_O_3_ particles are below 30 nm in size. As the dislocations move through the matrix, they are trapped by the nano-oxides. When a dislocation passes around the particle by combination of slip and climb, it can remain stuck to the back side of the nano-oxide and requires an increased force to detach from the oxide (see yellow arrows in Figure 15). In this way, the motion of the dislocations in the matrix is prohibited very effectively at temperatures above 1000 °C [209].

The FeAlOY ODS nanocomposite appears to be a very promising alloy for parts subjected to a long-term loading in oxidation environments at temperatures in the range of 1100–1300 °C. Concerning the related production technology, the hot consolidation of the canned FeAlOY powders is the key processing step. This operation determines the grain microstructure after secondary recrystallization and thus the resulting high-temperature mechanical properties. A good-quality and reproducible FeAlOY alloy fabricated in real industrial environment depends on a sufficiently wide temperature processing window available for hot consolidation, which would lead to the expected microstructures and properties [129]. These results showcase how future alloy can accelerate the discovery of revolutionary materials.

## 5. Austenitic ODS Alloys

In contrast to the ferritic ODS steels, the austenitic ODS variants received more attention only during the last decade [211]. One of the reasons for the increasing interest is a potentially better creep resistance due to their face-centered cubic (fcc) crystal structure. As compared to the more open bcc lattices, the closed-packed fcc structure reduces diffusion rates in terms of self-diffusion as well as diffusion of substitutional solutes by at least two orders of magnitude [212]. Similarly, considerable attention is also paid to the fcc-based austenitic ODS high-entropy alloys (HEAs). Another reason is that the higher Cr content of the austenitic alloys further increases the corrosion resistance. Several austenitic ODS steels derived from the commercial AISI 304 [204,213,214,215,216], AISI 310 [196,203,217] and AISI 316 [153,218,219] have been investigated regarding their microstructural and mechanical properties. The best-known variants are listed in Table 4.

Figure 16 shows the typical microstructure of ODS austenitic steel, including finely distributed oxide particles [224].

The high strength of austenitic ODS alloys is primarily attributed to the very small grain size and the retardation of grain growth at higher operating temperatures. Added nanoparticles inhibit dislocation motion and grain boundary migration, which increases the thermal stability of the microstructure. The dragging force exerted from precipitates on grain boundaries and dislocations is strongly dependent on the particle (1) size, (2) number density, (3) uniform distribution, (4) interfacial energy and (5) thermal stability [225,226]. Fortunately, the required particle characteristics can be achieved via an optimized processing route based on powder metallurgy. Nevertheless, the MA of the austenitic ODS steels is difficult to control and ensure a reproducible outcome due to a number of reasons including (i) the batch character of the process, (ii) non-equilibrium state of the final powder and (iii) issues associated with sticking between the powder and the attritor parts.

As an example, Figure 17a shows surfaces of the pre-alloyed austenitic powder particles and Figure 17b that of the as-milled powders both having a rather narrow size distribution. We note that the milled austenitic powders have a rather large particle size as compared to the ferritic ones and that of pre-alloyed powders; compare Figure 17a,b. The irregular surfaces of the milled powder suggest that a high portion of deformation energy was stored inside the particles. The shape of the powder changed from the initial spherical shape to a circular disk-type lump after milling. During the milling process, the ball-to-ball, ball-to-powder and ball-to-wall collisions contribute to the compression and shear deformation of the mechanically alloyed powders. In addition, the incidental and frequent rubbing of the particles lead to their flattening, fracture and welding, processes which form large particles due to the good plasticity of the pre-alloyed austenitic steel powders [204].

Indeed, the processing of the austenitic ODS steels by MA was shown to be challenging, mainly due to the formation or the presence of the very ductile fcc phase. Hence, the powder tends to stick to the container walls and milling balls and often so-called process-control agents (PCA) like alcohols have to be added to increase the powder yield [203,218,228]. These measures definitely decrease the production efficiency and alternate the chemical composition of the powder [168,227,229]. In the work by Gräning [230], usage of ZrO_2_ balls should be avoided to produce ODS steels due to the negative influence of abrasion, regardless of the higher production yield. ZrO_2_ debris ruins the mechanical properties, especially the ductility. Despite the sequestering of Zr in nano-sized precipitates, the presence of a number of big shards of ZrO_2_ in the alloy outweighs the benefits. Cryogenic MA represents a viable alternative with a drawback related to high production costs. In case of mechanically alloyed bcc steels, Ressel et al. [231] as well as Alinger et al. [232] provided evidence that yttria dissolves due to defects during milling and explained the role of vacancies in the dissolution process. The yttria-containing phases later reprecipitate into finely dispersed nanoparticles during the powder consolidation steps. Contrarily, Phaniraj et al. [5] did not confirm this scenario for the fcc austenitic steels, instead he concluded that yttria particles remain finely dispersed in the matrix [132]. As a result of the repeated fracturing and cold welding of the powders during MA, severe plastic deformation takes place in the powder particles [168].

A limited number of studies highlighted the fact that MA technology is rather inefficient as far as a yield of milled austenitic ODS powder is concerned [101,114,233]. The adhesion of ductile powders to the attritor parts during mechanical alloying is a well-known problem, which was already reported for aluminum and was tackled by the addition of stearic acid, as a carbon-rich PCA [234]. Applying the same procedure to the austenitic ODS steels helped to solve the issue, but additional studies are needed to rule out the detrimental impact of the processing route on the microstructure and mechanical properties of the final product [228].

Kim et al. [224] reported on the MA-based processing of austenitic ODS steel with a composition close to AISI 316L. Their ODS alloy attained an ultimate tensile strength of 660 MPa at room temperature. Expected Y-Ti-O-based complex oxides were not found. Instead, Y_2_Si_2_O_7_ and TiO_2_ particles with sizes in a range of hundred nanometers were observed in the steel matrix. One can expect that the chemical composition of the oxides results from a lack of Y and Ti and an increased amount of oxygen picked up by the oxidized powder surfaces. In order to improve the oxide characteristics, Si was used as the oxide-forming element. Subsequent cold rolling and secondary recrystallization yielded an isotropic grain structure and provided adequate tensile strength and ductility at both room temperature and 700 °C.

Zhou et al. also investigated ODS austenite steels and reported the ultimate tensile strength of 1000 MPa for the alloy containing Y-Ti-Si-O complex oxides in the 20 nm size range [204]. Follow up studies provided evidence that the addition of Zr and Hf to the PNC316 austenitic ODS steel annealed at 1200 °C further reduces the average size of the complex oxides from 14 nm down to 6 nm [235]. These complex oxides exhibit faceted morphology and an anion-deficient fluorite Y_2_Hf_2_O_7_-type crystal structure [236]. Zhou et al. compared the room temperature tensile strength of the base AISI304 austenitic steel and its HIPed ODS variant and showed that 17 nm sized Y-Ti-Si oxides promote the strength from 300 MPa to 940 MPa [214,216]. The high-number density of nano-oxides in austenitic ODS steels results in about a threefold increase in the yield and ultimate tensile strength at temperatures above 700 °C [204,216]. However, austenitic ODS steels exhibit low ductility in a temperature range of 600–900 °C, likely due to the formation of σ-phase (FeCr) that causes brittleness [214]. Moreover, during prolonged exposures at high temperatures, austenitic steels suffer from embrittlement due to the nucleation of Cr-rich (α′), Fe-rich (α), ordered FeCr and Laves phases [237].

Yingli Xu [216] reported on another austenitic 304 ODS type steel prepared by MA and then consolidated using hot isostatic pressing (HIP). Tungsten was alloyed in the ODS steel to improve the tensile properties at high temperatures. It was found that the oxide particles were very stable during annealing at 900 °C, although the grain size increased by a factor of two during annealing. Impurity atoms Al and Ca increased the oxide particle size significantly. The yield strength of the as-HIPed ODS alloy were significantly improved compared to 304 austenitic steel, but the ductility was degraded by coarse oxide particles.

The nickel-based ODS alloy MA 754 is strengthened by yttria additions. It has a melting temperature of 1400 °C and comparatively high yield strength at elevated temperatures. As the first mechanically alloyed Ni–Cr ODS alloy, it is used as a turbine vane and blade material in advanced jet engines. Its columnar grain structure, which is highly desirable for elevated temperature applications, is formed during isothermal annealing [238]. This microstructural evolution has been attributed to second-phase particles arranged into linear structures during hot rolling. The linear particle distribution inhibits grain growth perpendicular to the lines [239].

Super304H austenitic heat-resistant steel is frequently selected for high-temperature applications. The steel is based on 18Cr-8Ni stainless steel alloyed mainly with about 3 wt.% Cu and a small amount of Nb [213]. It was used as the base material which was strengthened by additions of 0.35 wt.% Y_2_O_3_ to the austenitic matrix. MA and HIP were applied as key steps during mixing and consolidation of the pre-alloyed matrix powder and the Y_2_O_3_ particles. The final ODS steel exhibits high tensile strength as tested at RT and 650 °C.

Although Fe–Cr ODS ferritic/martensitic steels have been developed primarily for ODS steels used in nuclear applications [88,114,240], a certain type of Fe–Cr–Ni ODS austenitic steel (316 L) has been developed as a high-temperature component because of its lower creep and diffusion rates [214,216,218]. In addition, its non-magnetism can be advantageous if the ferromagnetism of the ferritic/martensitic steels is prohibited in magnetic confinement-type fusion reactors and can be free from the huge magnetic force caused by the coils, especially the superconductor coils. As such, type-316L austenitic stainless steel has been selected as the principal structural material in the ITER [241].

Austenitic ODS alloys have not been considered as structural candidates for a while, since early neutron irradiation without exception showed the occurrence of severe swelling [242,243]. It is thought that adding oxide nanoparticles into austenitic steels may refine the microstructures and increase the quantity of interfaces, which will likely mitigate swelling. Thus to develop ODS austenitic steels and test their irradiation properties is of great interest. Nevertheless, irradiation studies on ODS austenitic steels are very limited so far [4,220,244].

In a study [220], cavities formed at a low dose of 0.1 dpa, which means helium is essential in cavity nucleation in ODS 316 steel at 500 °C. Y–Ti–O nanoparticles, grain boundaries and twin boundaries act as strong traps for cavities, which might mean that adding oxide particles can improve the radiation resistance. Helium also enhances the growth of interstitial-type defects. The Y–Ti–O particles in ODS 316 austenitic steel were unstable during irradiation. Figure 18 compares cavity microstructures after irradiation to 5 dpa with different helium injection rates. During the sole heavy ion irradiation, no cavities were found. With 20 appm/dpa helium, a few cavities were observed within the grain interior of sizes approximately 2–4 nm. With 200 appm/dpa helium, a high density of cavities was evenly distributed within the material.

Very important investigations were performed on the corrosive behavior of the ODS austenitic alloys in supercritical water (SCW). A supercritical water reactor (SCWR) is one of the most promising advanced concepts in generation IV reactor technology. Water at temperatures above 374.15 °C and a pressure higher than 22.1 MPa is in a supercritical state and behaves as a dense gas exhibiting properties significantly different from that of liquid water below the critical point [245,246]. At the critical point, the specific enthalpy of water increases by about 20% and the SCW behaves as a single phase [247]. The SCWR with a high thermal efficiency uses SCW as a coolant [248,249]. However, the SCW environment is extremely corrosive and this brings great challenges to the development and production of cladding materials. Austenitic 316 SS steel is used worldwide as a structural material in primary water and boiling water reactors due to its high corrosion and stress corrosion cracking resistance [250]. FM ODS steels were originally intended as fuel cladding materials for advanced SCWR due to their good mechanical strength and irradiation resistance in high-temperature nuclear environments [251,252,253]. However, the bcc crystalline structure of low Cr FM ODS steels exhibits poor corrosion resistance, while high Cr FM ODS steels show low ductility in SCW environments [251,253].

Most of the prospective fcc alloys show a much lower corrosion rate compared to bcc FM steels [254]. However, their mechanical strength significantly decreases at temperatures of about 600 °C [255]. The maximum fuel cladding hot spot temperature of 600 °C is typical in the normal SCWR operation mode with an outlet coolant temperature of 500–510 °C [254]. Thus, particle strengthening of 310 austenitic stainless steel would be required should the steel be considered for fuel-cladding material in SCWR. The ODS design is a potential way to increase both the mechanical strength and resistance to irradiation damage [62,63,251,252,256]. According to Shen [221], ODS 310 steel exhibits excellent corrosion resistance in SCW given by a solid-state growth mechanism of surface oxide layer controlled by ion diffusion.

Alumina-forming austenitic (AFA) stainless steels, where the protective layer is formed by alumina, were originally developed to improve the creep resistance of austenitic steels. The Oak Ridge National Laboratory [257] ran an extensive research program focused on the AFA steels. Based on the systematic studies, an optimum chemical composition of Fe-(20-25)Ni-(12-15)Cr-(3-4)Al-(1-3)Nb (wt.%) was recommended as the outcome. In the future, the AFA ODS alloys could become a promising material for fuel cladding in SCWR.

## 6. Conclusions

Research focused on steels for future nuclear power plants is aimed at alloys permitting the conversion of fusion or fission energy under extreme conditions in terms of temperature, mechanical stress, radiation and corrosion. In order to fulfil the requirements, mechanical alloying (MA) technology was optimized over the past few decades to enable the formation of nano-scale oxides dispersed uniformly in an alloy matrix. These chemically stable nano-particles efficiently block grain coarsening and, at the same time, they limit the welling-based damage of the components. This review provided a survey of austenitic, ferritic and ferritic/martensitic matrices by focusing on their, advantages, disadvantages and differences. The results can be summarized as follows:Ferritic ODS steel:Advantages:
-Bcc crystal structure of the matrix;-Higher operational temperature;-Low swelling;-High Cr content in bcc alloys increases the corrosion resistance;-Good oxidation resistance.
Disadvantages:
-Anisotropic mechanical properties;-The ferromagnetism;-Poor fracture toughness.
Ferritic–martensitic (FM) ODS steel:Advantages:
-Excellent creep resistance up to 800 °C as a consequence of a fine dispersion of insoluble oxide particles;-Large grain size and a high grain aspect ratio;-Nearly isotropic properties after heat treatment;-Scalable fabrication.
Disadvantages:
-Limited long-term applications at temperatures below 700 °C due to embrittlement associated with a precipitation of intermetallic phases;-The ferromagnetism;-Low oxidation resistance at high temperatures;-Poor fracture toughness.
Austenitic ODS steel:Advantages:
-Fcc crystal structure of the matrix;-Good ductility and toughness at high temperatures;-Better protection against corrosion;-Higher carbon solubility;-Non-magnetism;-High strength of austenitic ODS steels due to very small grain sizes and blocking of grain growth at high temperatures.
Disadvantages:
-The strength of the austenitic ODS steels is lower than ferritic counterparts at all temperatures;-Processing challenges associated with MA.

## Figures and Tables

**Figure 1 materials-17-03409-f001:**
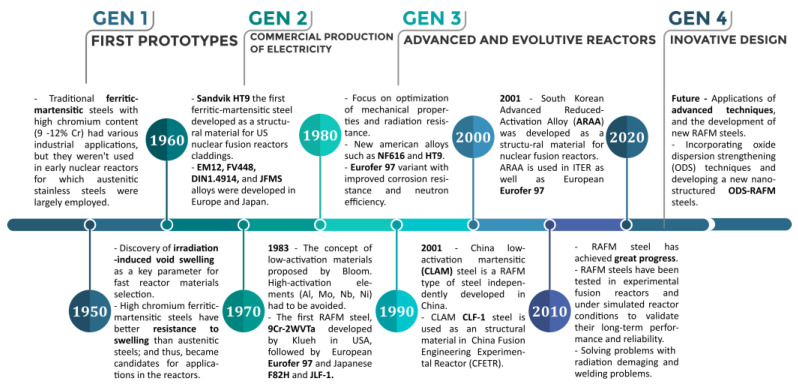
History of RAFM steels.

**Figure 2 materials-17-03409-f002:**
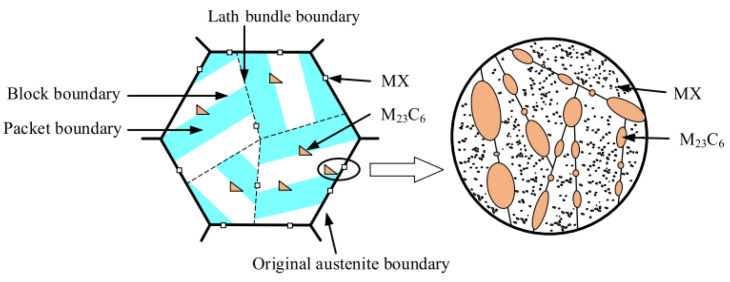
Schematic diagram of microstructure of tempered RAFM steel [30].

**Figure 3 materials-17-03409-f003:**
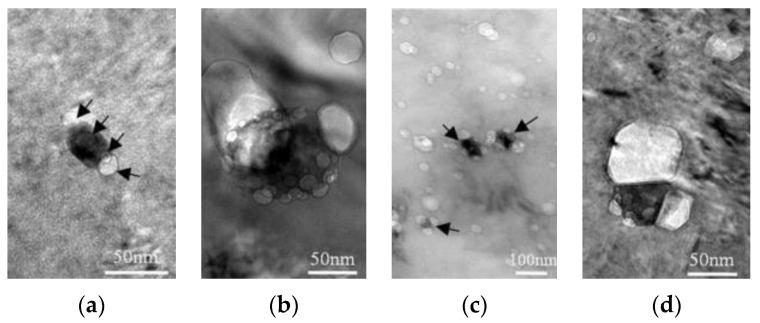
Preferential nucleation of cavities at interface of MC precipitates under different radiation doses (**a**) 2 dpa; (**b**) 6 dpa; (**c**) 9 dpa; (**d**) 18 dpa [30,45].

**Figure 4 materials-17-03409-f004:**
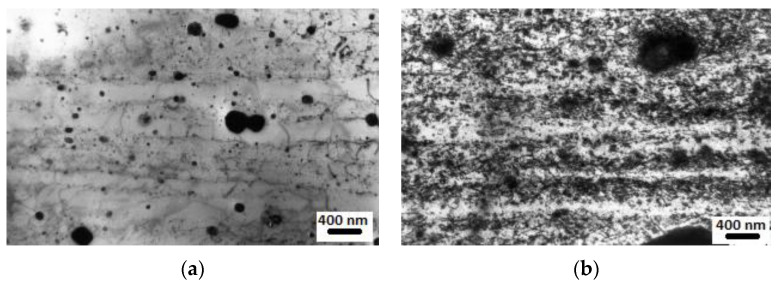
Oxide-free strips in the ODS ferritic steels before (**a**) and after irradiation and (**b**) at 435 °C—54.8 dpa [92].

**Figure 5 materials-17-03409-f005:**
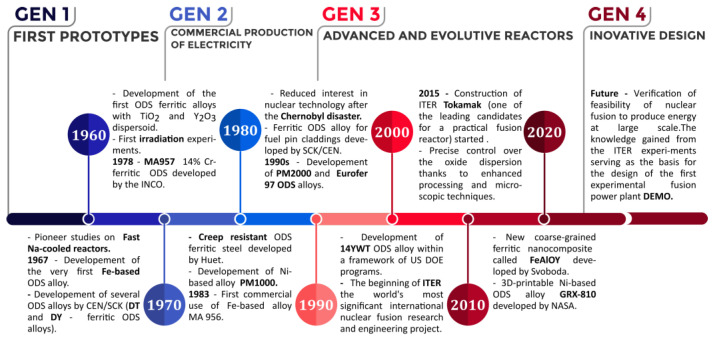
Origins and development of ODS steels.

**Figure 6 materials-17-03409-f006:**
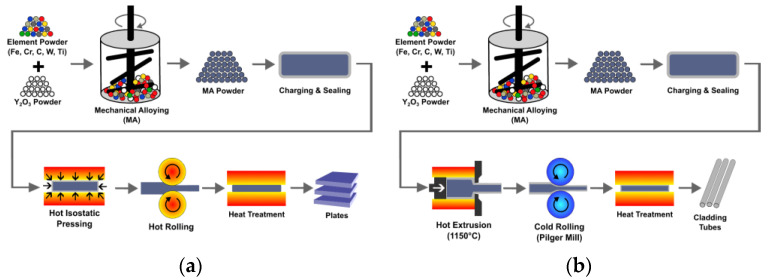
Schematic representation of ODS steel processing by powder metallurgy process, (**a**) manufacturing process of ODS steel plates and (**b**) manufacturing process of ODS steel cladding tubes.

**Figure 7 materials-17-03409-f007:**
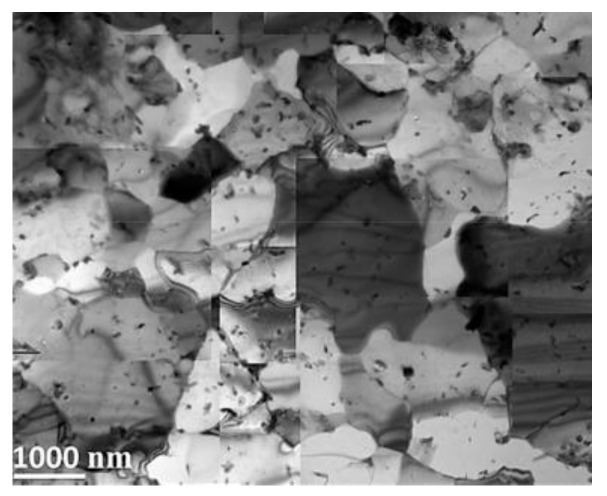
TEM bright field micrographs showing ODS Fe9Cr steel. Ferritic matrix with Cr carbides and low dislocation density [159].

**Figure 8 materials-17-03409-f008:**
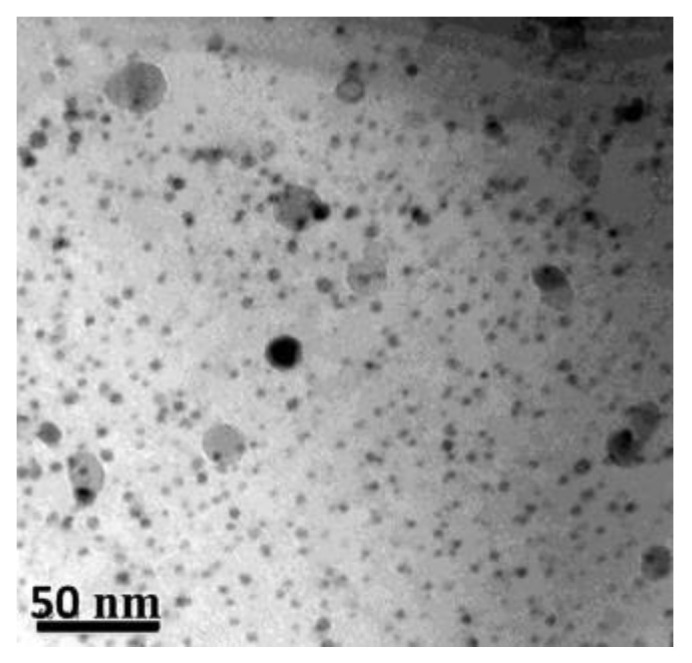
TEM bright filed micrograph of the heat-treated ODS Fe9Cr steel revealing region with uniform particle distribution [159].

**Figure 9 materials-17-03409-f009:**
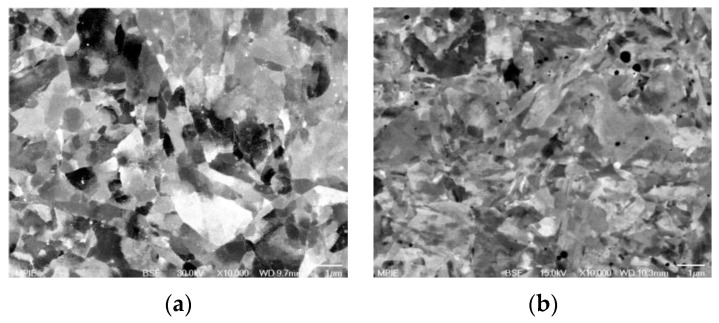
SEM BSE images showing microstructure of steels in the as-tempered condition (750 °C/2 h): (**a**) Eurofer-97, (**b**) ODS-Eurofer [35].

**Figure 10 materials-17-03409-f010:**
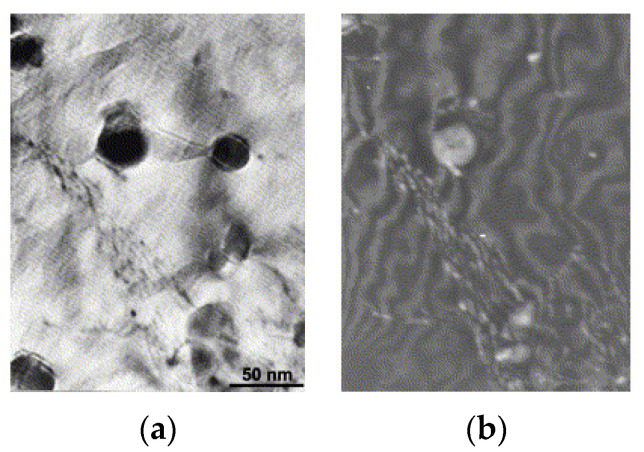
Microstructure of unirradiated 12Y1 ODS steel: bright (**a**) and dark field (**b**) TEM micrographs taken near a beam direction B ≈ 〈122〉. The weak-beam dark-field image is taken at g = 110 (g, 4 g) diffraction condition. Y_2_O_3_ precipitate particle sizes are in the range of a few tens of nanometers in diameter [107].

**Figure 11 materials-17-03409-f011:**
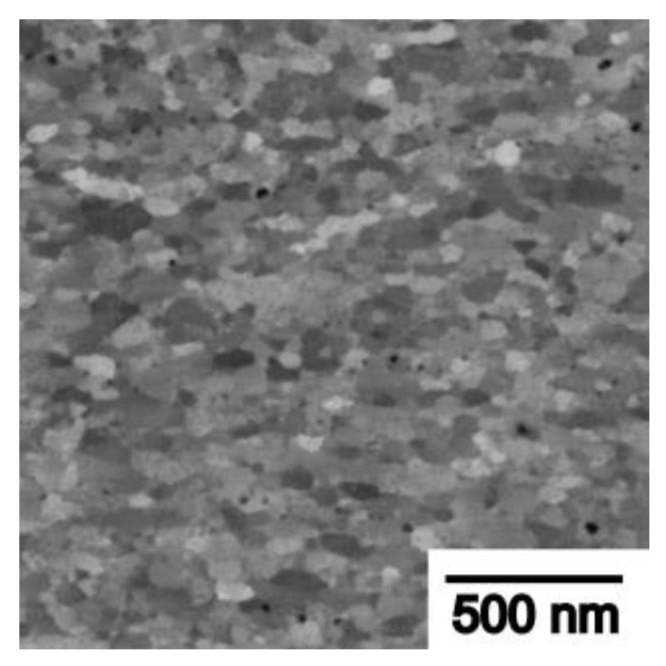
The typical microstructure of the consolidated FeAlOY [202].

**Figure 12 materials-17-03409-f012:**
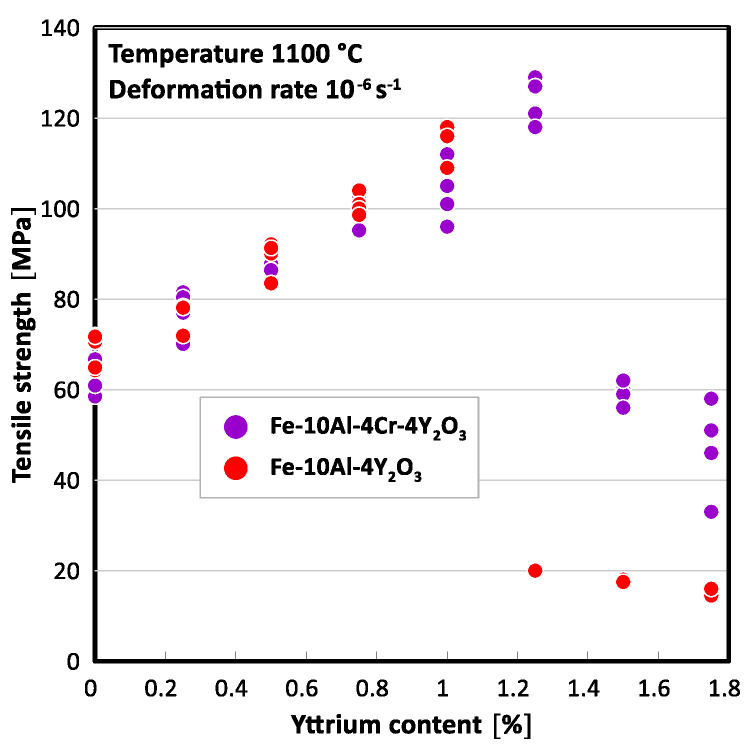
Tensile strengths of the Fe-10AlC4Y_2_O_3_ and Fe-10Al-4Cr-4Y_2_O_3_ ODS alloys and its changes with an increasing content of Y [205].

**Figure 13 materials-17-03409-f013:**
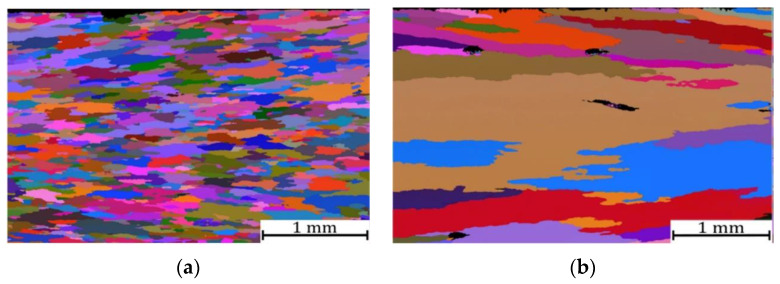
Colored EBSD maps of grain microstructure for Fe-10Al-4Y_2_O_3_ ODS alloys with addition of (**a**) 0% Y, (**b**) 1.25% Y (black areas are not recrystallized) [205].

**Figure 14 materials-17-03409-f014:**
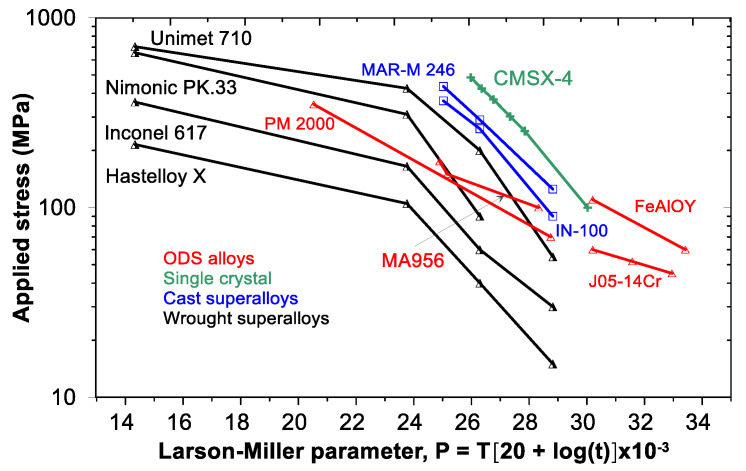
Applied stress vs. Larson-Miller parameter for selected the commercial and experimental ODS steels, superalloys and single crystal [109,205,206,207,208].

**Figure 15 materials-17-03409-f015:**
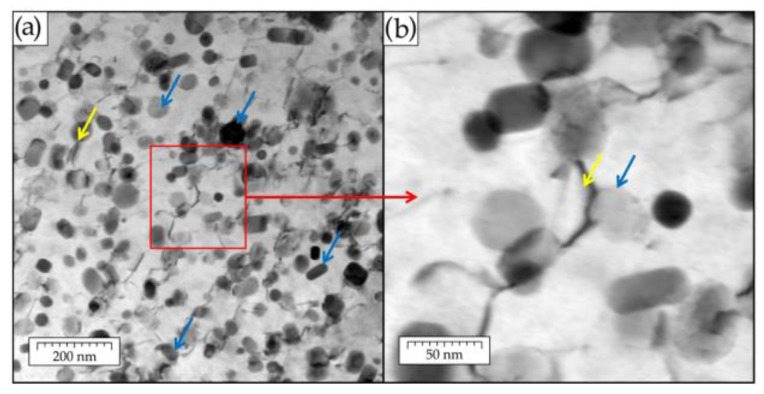
STEM micrographs (bright field) of the FeAlOY microstructure. Blue arrows show Y_2_O_3_ particles, yellow arrows show dislocations prohibited in their motion by nanooxides. Part (**b**) is a magnified area of (**a**) that is highlighted by the red square [210].

**Figure 16 materials-17-03409-f016:**
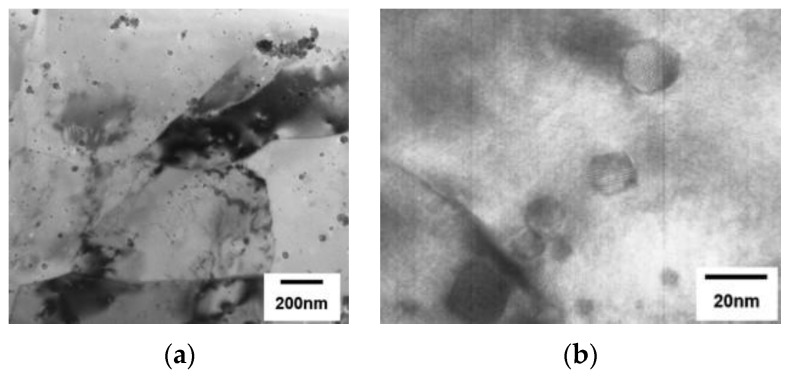
(**a**) Typical microstructure of ODS austenitic steel and (**b**) TEM micrograph showing faceted particles having a diameter of around 10 nm in the ODS austenitic steel [224].

**Figure 17 materials-17-03409-f017:**
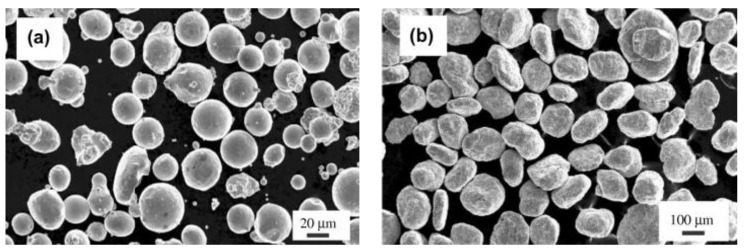
SEM images documenting a morphology of the powders; (**a**) pre-alloyed powders, (**b**) mechanically alloyed ODS powders [227].

**Figure 18 materials-17-03409-f018:**
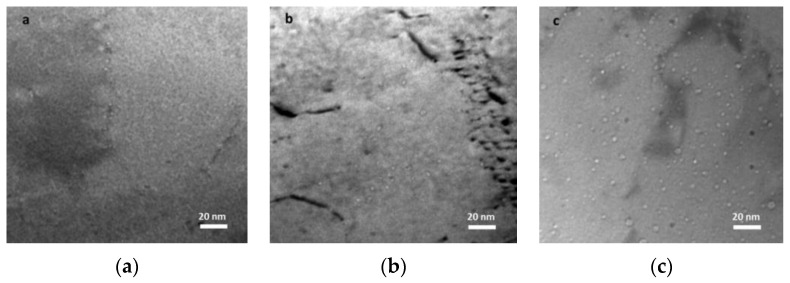
TEM bright field micrographs showing microstructures after irradiation to 5 dpa, with simultaneous injection of (**a**) no helium, (**b**) 20 appm/dpa helium and (**c**) 200 appm/dpa helium [220].

**Table 1 materials-17-03409-t001:** Chemical compositions and microstructures of typical RAFM steels [29,30,36].

RAFM Steel	Chemical Composition	Matrix Structure
Eurofer97	(8.5–9.5)Cr-1W-0.2V-0.1Ta-0.4Mn	Complete martensite
9Cr-2WVTa	9Cr-2W-0.18V-0.08Ta	Martensite
F82H	8Cr-2W-0.2V-0.004Ta-0.1C	Martensite
JLF-1	9Cr-2W-0.18V-0.08Ta	Ferrite
CLAM	9Cr-1.5W-0.5Mn-0.2V-0.15Ta-0.11C	Complete martensite
CLF-1	8.5Cr-1.5W-0.5Mn-0.3V-0.1Ta-0.1C	Complete martensite
BATMAN II	9Cr-1.5W-0.2Ti-0.25V-3.5Mn	Complete martensite
ARRA	9Cr-0.9W-0.18V-0.06Ta-0.09Si	Martensite

**Table 2 materials-17-03409-t002:** Decay time of main elements [30].

Elements	Activation	Decay Time
Li, Be, B, C, O, Si, S, P	Very low	14 days
Ti, V, Cr, Zr, W, Pb, Y	Low	30 days–5 years
Mn, Fe. Zn, Hf	High	10–30 years
Al, Ni, Cu, Nb. Mo, Sn	Very high	>100 years

**Table 3 materials-17-03409-t003:** Chemical compositions of some ferritic ODS alloys considered for nuclear applications.

ODS Steel	Fe	Cr	W	Ti	Mo	Mn	Al	C	Y_2_O_3_	Other
ODS Eurofer [35]	Bal.	9	1.1	-	-	0.4	-	0.11	0.3	0.2 V; 0.12 Ta
12Y1 [107]	Bal.	12.85	<0.01	0.03	0.03	0.04	0.007	0.045	0.25	0.03 Si; 0.24 Ni
12YW [107]	Bal.	12	3	-	-	-	-	-	0.3	-
12YWT [107]	Bal.	12	3	0.4	-	-	-	-	0.25	-
14YWT [77]	Bal.	14	3	0.4	-	-	-	-	0.3	-
Fe-18Cr1W [108]	Bal.	18	0.95	0.25	-	0.3	-	0.03	0.5	0.3 Si; 0.2 Ni
MA956 [109]	Bal.	20	-	04	-	-	4.5	-	0.5	
MA957 [110]	Bal.	14	-	0.9	0.3	-	0.03	0.01	0.25	
PM2000 [110]	Bal.	20	-	0.5	-	-	5.5	-	0.5	
C36M [111]	Bal.	12	-	-	2	-	6	-	0.03	0.2 Si
FeCrAl-ODS-1 [112]	Bal.	12	-	0.5	-	-	6	-	0.5	0.4 Zr
FeCrAl-ODS-2 [113]	Bal.	13	-	0.5	2	-	5	-	0.35	0.6 Zr
9CrODS [114]	Bal.	9	1.5	0.5	-	-	-	0.1	0.35	0.2 V
12CrODS [115]	Bal.	12	1.2	0.5	-	0.5	-	0.1	0.35	
14CrODS [116]	Bal.	14	2	0.5	-	-	-	-	0.35	
18CrODS [117]	Bal.	18	2.	0.5	4	-	4	-	0.35	

**Table 4 materials-17-03409-t004:** Chemical compositions of selected austenitic ODS steels.

ODS Steel	Fe	Cr	Ti	Mo	Mn	C	Y_2_O_3_	Ni	Si	Nb
316ODS [220]	Bal.	17	0.0.8	2.5	0.4	-	0.35	13	0.7	0.08
304ODS [214]	Bal.	18	0.5	1	-	-	0.35	8	0.15	-
310ODS [221]	Bal	25	0.2	2	-	0.05	0.35	20	-	-
MA754 [222]	1	20	0.5	-	-	0.05	0.6	77.55	-	-
ODS-800H [223]	41.6	21	-		0.83	0.07	-	Bal.	0.42	-

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
