# Peer review of "The Irradiation Effects in Ferritic, Ferritic–Martensitic and Austenitic Oxide Dispersion Strengthened Alloys: A Review"

_materials, 2024, doi:10.3390/ma17143409_

Round 1
Reviewer 1 Report
Comments and Suggestions for Authors
The authors have presented a comprehensive scientific review of the alloys used in the nuclear industry. The papers show the most important facts about the development of the new alloys and associated problems. The authors have deep knowledge of the topic. The paper does not contain any critical errors and thus does not need extensive corrections and improvements. The scientific topic concerning alloys for the nuclear industry is extremely large and not all aspects can be presented just in one paper, which is fully understandable. I would recommend adding references about research about other aspects of the material science for the nuclear industry (e.g. https://doi.org/10.1016/j.nucengdes.2013.06.018 )
Author Response
Thank you for your review, questions, comments and suggestions. This is highly beneficial during the final editing of the text.
It is certainly welcome to add other aspects of the material science for the nuclear industry to the review. I added the impact from your recommended paper by Tuček and joined the paper “Materials selection for nuclear application: Challenges and opportunities” by Hosemann. The changes are marked in yellow on pages 2 and 11 of the revised version.

Reviewer 2 Report
Comments and Suggestions for Authors
The paper reports an overview on the irradiation resistant alloys predominantly strengthened by oxide dispersions. The topic could be interesting for readers of Materials. The presentation and discussion could be slightly improved. On this basis, I recommend the publication after the following minor revisions:
- Figure 14 reports the Larson-Miller diagram. I suggest to better describe the findings that can be determined from the analysis of Larson-Miller diagram.
- Pag.22. The authors stated “STEM micrographs are presented in Fig. X”. “X” should be replaced with “15”.
- Pag. 28. The authors stated “This drop in strength is accompanied by a significant increase of ductility”. Figure 12. Could the authors add the data on the tensile parameter associated with the ductility?
Comments on the Quality of English LanguageMinor corrections are needed.
Author Response
Thank you for your review, questions, comments and suggestions. This is highly beneficial during the final editing of the text.
Figure 14: Based to your recommendation, description of the findings that can be determined from the analysis of Larson-Miller diagram are on the page 25.
Pag.22: The text error has now been corrected.
Pag. 28: Necessary data associated with the ductility are referred on page 24.

Reviewer 3 Report
Comments and Suggestions for Authors
This paper presents a comprehensive review of several Oxide Dispersion Strengthened (ODS) alloys, which are slated for potential publication in this journal post-revisions.
1. The graphical elements within this article suffer from inconsistent layout and labeling.
2. The extensive citation of over 200 articles juxtaposed with fewer than 20 images raises concerns regarding the professional presentation of this review.
3. Notably, Figure 8 exhibits duplicity, featuring identical images, which warrants clarification.
4. The inconsistent bolding of temperature descriptions within the article prompts inquiry into the rationale behind this discrepancy.
5. It is imperative to meticulously scrutinize the article's details; for instance, discrepancies exist wherein some numeric units feature leading spaces while others do not.
6. The article is beset by significant issues concerning logical coherence and relevance. For instance, while the title emphasizes the irradiation effect, its manifestation within the article remains inadequately elucidated. Substantive revisions are warranted to align the content with the purported focus.
Comments on the Quality of English LanguageThis paper presents a comprehensive review of several Oxide Dispersion Strengthened (ODS) alloys, which are slated for potential publication in this journal post-revisions.
1. The graphical elements within this article suffer from inconsistent layout and labeling.
2. The extensive citation of over 200 articles juxtaposed with fewer than 20 images raises concerns regarding the professional presentation of this review.
3. Notably, Figure 8 exhibits duplicity, featuring identical images, which warrants clarification.
4. The inconsistent bolding of temperature descriptions within the article prompts inquiry into the rationale behind this discrepancy.
5. It is imperative to meticulously scrutinize the article's details; for instance, discrepancies exist wherein some numeric units feature leading spaces while others do not.
6. The article is beset by significant issues concerning logical coherence and relevance. For instance, while the title emphasizes the irradiation effect, its manifestation within the article remains inadequately elucidated. Substantive revisions are warranted to align the content with the purported focus.
Author Response
Thank you for your review, questions, comments and suggestions. This is highly beneficial during the final editing of the text.
- The found errors in the text have been corrected.
- The authors set out to identify a professional and scientific approach by creating sufficient citation-based information with a respective number of illustrative figures. It was very difficult to select only the most essential figures, so that the reader would have a general overview. In our opinion, an appropriate equilibrium between the textual content and the visual elements has been achieved. The reader more interested in details must read the original papers referenced in an extensive amount.
- This error occurred when inserting an article into the Materials template. This error is corrected.
- This mistake occurred during formatting of article. Bolding of temperature should only be visible to authors, not to readers. Rest assured, this error will be corrected right away.
- The errors have been addressed and corrected.
- Based on your recommendation, the authors have attempted to modify the text so that the review has sufficient coherence and relevance.

Reviewer 4 Report
Comments and Suggestions for Authors
This manuscript presents an interesting and comprehensive review of the irradiation effects in ferritic, ferritic-martensitic, and austenitic oxide dispersion-strengthened alloys. The manuscript can be considered for publication after a minor revision. The main concerns are
1. The fourth paragraph on page 3 introduces the martensitic transformation and its effects. However, it is not closely related to the context of radiation swelling.
2. The statement of 'it improves mechanical properties (Hall–Petch relationship)' in the fourth paragraph on page 3 is not rigorous. Usually, the martensitic transformation will change the crystal structure. If the crystal structure changes, it should not be attributed to the Hall-Petch effect. The Hall-Petch effect refers to the effect of the grain size of the material on the strength and toughness under the same crystal structure. In addition, grain refinement will increase the strength of the material, but it will sacrifice the toughness of the material. Simply saying that the mechanical properties are improved is not rigorous enough. It is necessary to explain which one is more important for nuclear steel, strength or toughness.
3. The details of the steps for producing ODS alloys on page 13 are not relevant to the main purpose of the article and are not recommended to be listed.
4. Figure 8b is the same as Figure 8a. The lattice image of the Y2Ti2O7 nano-oxides mentioned in the article is not shown in Figure 8b.
5. In conclusion, the bcc and fcc crystal structures of the matrix, which is better? Why were they listed as an advantage for Ferritic ODS steel and Austenitic ODS steel, respectively?
6. It seems that not many papers published in the past ten years are listed. For review articles, recent research progress is very important.
Author Response
Thank you for your review, questions, comments and suggestions. This is highly beneficial during the final editing of the text.
- We are committed to correcting this error immediately.
- The role of the martensitic transformation consists in the fact, that it allows to transform the chemically homogenized austenite to supersaturated bcc lattice without a significant chemical heterogenization. Then the subsequent heat treatment allows tuning the microstructure by proper heat treatment leading to mechanical properties required by a given application. The martensitic transformation can be thus considered as one of the processing steps. This not rigorous assertion has been omitted.
- The producing steps are described quite shortly and could provide some basic information to readers interested in this topic.
- It is true, lattice image of the Y2Ti2O7 nano-oxides mentioned in the article is not shown in Figure 8b as well as Figure 8b is the same as Figure 8a. These errors have been identified and corrected.
- The aim of the article was not to find out which type of alloy is “better”, but to look more comprehensively at the nature of their matrix when exposed to irradiation and to describe their advantages and disadvantages. When creating new reactor designs, this information can help in identifying the most suitable materials.
- It is true that some of the articles in the review are older (for example from 1967). But since an overall timeline of the development of RAFM and ODS steels was being established, it was necessary to include these references as well. As far as older publications are concerned, these are reputable articles with a large number of citations. From the most recent (older than 5 years) references more than 30 references are listed.

Round 2
Reviewer 3 Report
Comments and Suggestions for Authors
The authors made a good response to the reviewer's comments, and the manuscript is therefore recommended for acceptance.
Comments on the Quality of English LanguageThe authors made a good response to the reviewer's comments, and the manuscript is therefore recommended for acceptance.